# BOOSTING LATENT DIFFUSION WITH PERCEPTUAL OBJECTIVES

**Tariq Berrada Ifriqi**[1,2], **Pietro Astolfi**[1], **Melissa Hall**[1], **Marton Havasi**[1], **Yohann Benchetrit**[1],
**Adriana Romero-Soriano**[1,3,4,5], **Karteek Alahari**[2], **Michal Drozdzal**[1], **Jakob Verbeek**[1]

[1] FAIR at Meta, [2] Univ. Grenoble Alpes, Inria, CNRS, Grenoble INP, LJK, France
[3] McGill University, [4] Mila, Quebec AI institute, [5] Canada CIFAR AI chair
`tariqberrada@meta.com`

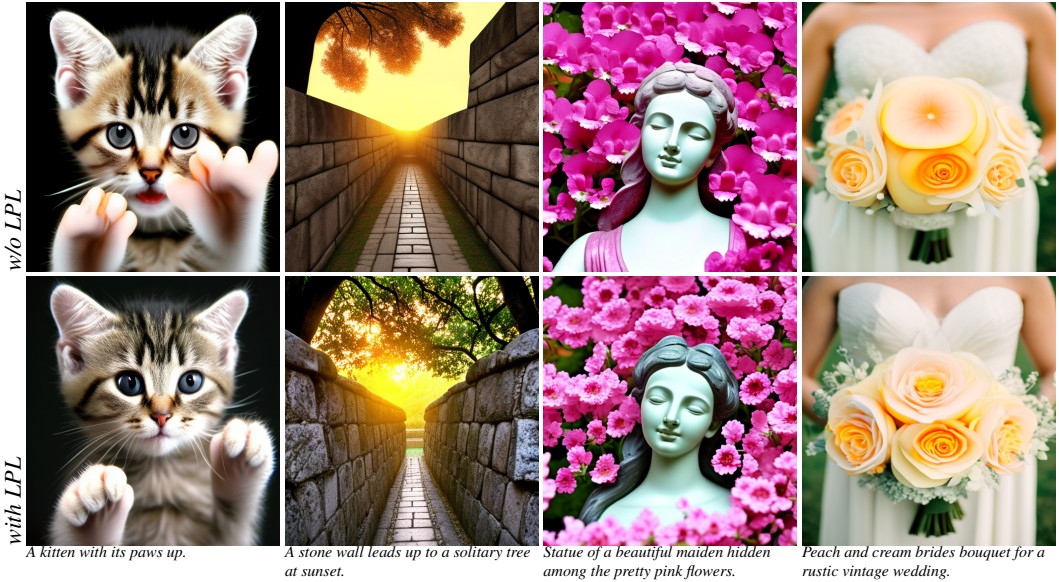

Figure 1: **Samples from models trained with and without our latent perceptual loss on CC12M.**
Samples from our model with latent perceptual loss (bottom) have more detail and realistic textures.

## ABSTRACT

Latent diffusion models (LDMs) power state-of-the-art high-resolution generative image models. LDMs learn the data distribution in the latent space of an autoencoder (AE) and produce images by mapping the generated latents into RGB image space using the AE decoder. While this approach allows for efficient model training and sampling, it induces a disconnect between the training of the diffusion model and the decoder, resulting in a loss of detail in the generated images. To remediate this disconnect, we propose to leverage the internal features of the decoder to define a *latent perceptual loss* (LPL). This loss encourages the models to create sharper and more realistic images. Our loss can be seamlessly integrated with common autoencoders used in latent diffusion models, and can be applied to different generative modeling paradigms such as DDPM with epsilon and velocity prediction, as well as flow matching. Extensive experiments with models trained on three datasets at 256 and 512 resolution show improved quantitative – with boosts between 6% and 20% in FID – and qualitative results when using our perceptual loss.

## 1 INTRODUCTION

Latent diffusion models (LDMs) (Rombach et al., 2022) have enabled considerable advances in image generation, and elevated the problem of generative image modeling to a level where it has

become available as a technology to the public. A critical part to this success is to define the generative model in the latent space of an autoencoder (AE), which reduces the resolution of the representation over which the model is defined, thereby making it possible to scale diffusion methods to larger datasets, resolutions, and architectures than original pixel-based diffusion models (Dhariwal & Nichol, 2021; Sohl-Dickstein et al., 2015).

To train an LDM, all images are first projected into a latent space with the encoder of a pre-trained autoencoder, and then, the diffusion model is optimized directly in the latent space. Note that when learning the diffusion model the AE decoder is not used – the diffusion model does not receive any training feedback that would ensure that all latent values reachable by the diffusion process decode to a high quality image. This training procedure leads to a disconnect between the diffusion model and the AE decoder, prompting the LDM to produce low quality images that oftentimes lack high frequency image components. Moreover, we note that the latent spaces of pre-trained LDM's autoencoders tend to be highly irregular, in the sense that small changes in the latent space can lead to large changes in the generated images, further exacerbating the autoencoder-diffusion disconnect problem.

In this work, we propose to alleviate this autoencoder-diffusion disconnect and propose to include the AE decoder in the training objective of LDM. In particular, we introduce latent perceptual loss (LPL) that acts on the decoder's intermediate features to enrich the training signal of LDM. This is similar to the use of perceptual losses for image-to-image translation tasks (Johnson et al., 2016; Zhang et al., 2018), but we apply this idea in the context of generative modeling and use the feature space of the pre-trained AE decoder rather than that of an external pre-trained discriminative network. Our latent perceptual loss results in sharper and more realistic images, and leads to better structural consistency than the baseline – see Figure 1. We validate LPL on three datasets of different sizes – the commonly used datasets ImageNet-1k (1M data points) and CC12M (12M data points), and additionally a private dataset S320M (320M data points) – as well as three generative models formulation – DDPM (Ho et al., 2020) with velocity and epsilon prediction, and conditional flow matching model (Lipman et al., 2023). In our experiments, we report standard image generative model metrics – such as FID (Heusel et al., 2017), CLIPScore (Hessel et al., 2021), as well as Precision and Recall (Sajjadi et al., 2018; Kynkäänniemi et al., 2019). Our experiments show that the use of LPL leads to consistent performance boosts between 6% and 20% in terms of FID. Our qualitative analysis further highlights the benefits of LPL, showing images that are sharp and contain high-frequency image details.

In summary, our contributions are:
- We identify a slight disconnect between latent and pixel-space diffusion which can lead to suboptimal results when training latent diffusion and flow models.
- We propose the *latent perceptual loss (LPL)*, a perceptual loss variant leveraging the intermediate feature representation of the autoencoder's decoder.
- We present extensive experimental results on the ImageNet-1k, CC12M, and S320M datasets, demonstrating the benefits of LPL in boosting the model's quality by 6% to 20% in terms of FID.
- We show that LPL is effective for a variety of generative model formulations including DDPM and conditional flow matching approaches.

## 2 RELATED WORK

**Diffusion models.** The generative modeling landscape has been significantly impacted by diffusion models, surpassing previous state-of-the-art GAN-based methods (Brock et al., 2019; Karras et al., 2019; 2020; 2021). Diffusion models offer advantages such as more stable training and better scalability, and were successfully applied to a wide range of applications, including image generation (Chen et al., 2024; Ho et al., 2020), video generation (Ho et al., 2022b; Singer et al., 2023), music generation (Levy et al., 2023; San Roman et al., 2023), and text generation (Wu et al., 2023). Various improvements of the framework have been proposed, including different schedulers (Lin et al., 2024; Hang & Gu, 2024), loss weights (Choi et al., 2022; Hang et al., 2023b), and more recently generalizations of the framework with flow matching (Lipman et al., 2023). In our work we evaluate the use of our latent perceptual loss in three different training paradigms: DDPM under noise and velocity prediction, as well as flow-based training with the optimal transport path.

**Latent diffusion.** Due to the iterative nature of the reverse diffusion process, training and sampling diffusion models is computationally demanding, in particular at high resolution. Different approaches have been explored to generate high-resolution content. For example, Ho et al. (2022a) used a cascaded approach to progressively add high-resolution details, by conditioning on previously generated lower resolution images. A more widely adopted approach is to define the generative model in the latent space induced by a pretrained autoencoder (Rombach et al., 2022), as previously explored for discrete autoregressive generative models (Esser et al., 2021). Different architectures have been explored to implement diffusion models in the latent space, including convolutional UNet-based architectures (Rombach et al., 2022; Podell et al., 2024), and more recently transformer-based ones (Peebles & Xie, 2023; Chen et al., 2024; Gao et al., 2023; Esser et al., 2024) which show better scaling performance. Working in a lower-resolution latent space accelerates training and inference, but training models using a loss defined in the latent space also deprives them from matching high-frequency details from the training data distribution. Earlier approaches to address this problem include the use of a refiner model (Podell et al., 2024), which consists of a second diffusion model trained on high-resolution high-quality data that is used to noise and denoise the initial latents, similar to how SDEdit works for image editing (Meng et al., 2022). Our latent perceptual loss addresses this issue in an orthogonal manner by introducing a loss defined across different layers of the AE decoder in the latter stages of the training process. Our approach avoids the necessity of training on specialized curated data (Dai et al., 2023), and does not increase the computational cost of inference.

**Perceptual losses.** The use of internal features of a fixed, pre-trained deep neural network to compare images or image distributions has become common practice as they have been found to correlate to some extent with human judgement of similarity (Johnson et al., 2016; Zhang et al., 2018). An example of this is the widely used Fréchet Inception Distance (FID) to assess generative image models (Heusel et al., 2017). Such "perceptual" distances have also been found to be effective as a loss to train networks for image-to-image tasks and boost image quality as compared to using simple $\ell_1$ or $\ell_2$ reconstruction losses. They have been used to train autoencoders (Esser et al., 2021), models for semantic image synthesis (Isola et al., 2017; Berrada et al., 2024) and super-resolution (Suvorov et al., 2022; Jo et al., 2020), and to assess the sample diversity of generative image models (Schönfeld et al., 2021; Astolfi et al., 2024). In addition, recent works propose variants that do not require pretrained image backbones (Amir & Weiss, 2021; Czolbe et al., 2020; Veeramacheneni et al., 2023). An et al. (2024); Song et al. (2023) employed LPIPS as metric function in pixel space to train cascaded diffusion models and consistency models, respectively. More closely related to our work, Kang et al. (2024) used a perceptual loss (*E-LatentLPIPS*) defined in latent space to distill LDMs to conditional GANs, but used a separate image classification network trained over latents rather than the autoencoder's decoder to obtain the features for this loss. Lin & Yang (2024) added a perceptual loss (*Self-Perceptual*) to train LDMs that is defined over the features of the denoiser network in the case of UNet architectures. However, they found this loss to be detrimental when using classifier-free guidance for inference. In summary, compared to prior work on perceptual losses, our work is different in that (i) LPL is defined over the features of the decoder – that maps from latent space to RGB pixel space, rather than using a network that takes RGB images as input – and (ii) our LPL can be regardlessly applied to train both latent diffusion and flow models and has no architecture constraint in the denoiser model.

## 3 USING THE LATENT DECODER TO DEFINE A PERCEPTUAL LOSS

In this section, we analyze the impact of the decoder-diffusion disconnect on the LDM training, and then, we follow with the definition of our latent perceptual loss.

### 3.1 LATENT DIFFUSION AND THE $\ell_2$ OBJECTIVE

We use $F_\beta$ to refer to an autoencoder that consists of two modules. The encoder, $F_\beta^e$, maps RGB images $\mathbf{x}_0 \in \mathbb{R}^{H \times W \times 3}$ to a latent representation $\mathbf{z}_0 \in \mathbb{R}^{H/d \times W/d \times C}$, where $d$ is the spatial downscaling factor, and $C$ the channel dimension of the autoencoder. The decoder, $F_\beta^d$, maps from the latent space to the RGB image space. In LDM, the diffusion model, $D_\Theta$, with parameters $\Theta$ is defined over the latent representation of the autoencoder. We follow a typical setting, see *e.g.* (Peebles & Xie, 2023; Chen et al., 2024; Rombach et al., 2022), where we use a fixed pre-trained autoencoder with a downsampling factor of $d = 8$ and a channel capacity of $C = 4$.

**Training Objective.** The diffusion formulation results in an objective function that is a lower-bound on the log-likelihood of the data. In the DDPM paradigm (Ho et al., 2020), the variational lower bound can be expressed as a sum of denoising score matching losses (Vincent, 2011), and the objective function can be written as $\mathcal{L} = \sum_t \mathcal{L}_t$, where $\mathcal{L}_t = \mathcal{D}_{\text{KL}}\left[q\left(\mathbf{x}_{t-1}|\mathbf{x}_t, \mathbf{x}_0\right)||p_\theta\left(\mathbf{x}_{t-1}|\mathbf{x}_t\right)\right] = \mathbb{E}_{\mathbf{x}_0, \boldsymbol{\epsilon}, t}\left[\frac{\beta_t}{(1-\beta_t)(1-\alpha_t)} \cdot \|\boldsymbol{\epsilon} - \epsilon_\theta(\mathbf{x}_t, t)\|^2\right]$. Ho et al. (2020) observed that disregarding the time step specific weighting resulted in improved sample quality, and introduced a simplified noise reconstruction objective, known as epsilon prediction, where the objective is the average of the MSE loss between the predicted noise and the noise vector added to the image, $\mathcal{L}_{\text{simple}} = \sum_t \mathbb{E}_{\mathbf{x}_0, \boldsymbol{\epsilon}, t}\left[\|\boldsymbol{\epsilon} - \epsilon_\theta(\mathbf{x}_t, t)\|^2\right]$. The underlying idea is that the better the noise estimation, the better the final sample quality. An equivalent way to interpret this objective is through reparameterization of the target to the original latents, $\mathcal{L}_{\text{simple}} = \sum_t \lambda_t \cdot \mathbb{E}_{\mathbf{x}_0, \boldsymbol{\epsilon}, t}\left[\|\mathbf{x}_0 - \hat{x}_0(\mathbf{x}_t, t; \theta)\|^2\right]$, where $\hat{x}_0(\mathbf{x}_t, t; \theta) = (\mathbf{x}_t - \sigma_t \hat{\boldsymbol{\epsilon}}_t)/\alpha_t$ and $\lambda_t = 1/\sigma_t^2$.

We note that the presence of $\ell_2$ in the LDM objective has two important implications. First, the $\ell_2$ norm treats all pixels in the latents as equally important and disregards the downstream structure induced by the decoder whose objective is to reconstruct the image from its latents. This is problematic because the autoencoder's latent space has a non-uniform structure and is not equally influenced by the different pixels in the latent code. Thus, optimizing the $\ell_2$ distance in the diffusion model latent space could be different from optimizing the perceptual distance between images. Second, while an $\ell_2$ objective is theoretically justified in the original DDPM formulation, generative models trained with an $\ell_2$ reconstruction objective have been observed to produce blurry images, as is the case *e.g.* for VAE models (Kingma & Welling, 2014). Such an effect can lead to exposure bias in diffusion (*i.e.* denoiser input mismatch between training and sampling), which has been studied through the lens of the sampling algorithm but not the training objective Ning et al. (2024). The problem of blurry images due to $\ell_2$ reconstruction losses has been addressed through the use of perceptual losses such as LPIPS (Zhang et al., 2018), which provide a significant boost to the image quality in settings such as autoencoding (Esser et al., 2021), super-resolution Ledig et al. (2017) and image-to-image generative models (Isola et al., 2017; Park et al., 2019).

To address these two implications of the latent $\ell_2$ optimization, we design a new loss, namely Latent Perceptual Loss (LPL), that directly tackles the mismatch between the structures of the latent and the pixel space by matching the target and the predicted latents when partially decoded with the AE decoder. Compared to existing perceptual losses, e.g., LPIPS, that usually rely on features from pre-trained classifier networks, the use of AE decoder features is not standard. However, for LDMs training, the AE decoder provides a perceptually meaningful signal as the intermediate features of the decoder, being closer to pixel space, have a structure more similar to the pixel space. Moreover, these features have higher resolution than the diffusion latent features, which helps in modeling high-level details that reduce blurriness of the predicted image.

## 3.2 Latent Perceptual Loss

We propose a loss function that operates on the features at different depths in the autoencoder's decoder, $F_\beta^d$. Let $\mathbf{z}_t = \alpha_t \mathbf{z}_0 + \sigma_t \boldsymbol{\epsilon}_t$ be a noisy sample in the diffusion model latent space at time $t$, and $\hat{\mathbf{z}}_0 = (\mathbf{z}_t - \sigma_t D(\mathbf{z}_t, t; \Theta))/\alpha_t$ the corresponding estimated noise-free latent at time $t = 0$.

To enhance the accuracy of the reconstruction process in image space, we propose augmenting the diffusion objective with a penalty term that encourages the predicted forward process distribution to match the reconstructions $\hat{\mathbf{z}}_0$ at a lower level of compression. This is achieved by introducing an image-level projection penalty on the forward process, which can be expressed as:

$$\mathcal{L}_{t-1}^{\text{pen}} = \mathbb{E}_q\left[D_{\text{KL}}\left(q(\mathbf{x}_{t-1}|\mathbf{x}_t, \mathbf{x}_0) \| p_\Theta(\hat{\mathbf{x}}_{t-1}|\hat{\mathbf{x}}_t)\right)\right]. \tag{1}$$

In Appendix A.1, we show that such a penalty, under certain conditions, can be approximated by computing a pairwise distance between the outputs of the projection decoder, which maps latents into samples in image-space. Guided by this result, our loss term should resemble a reconstruction term between the original image and the predicted/denoised image obtained from the decoder of the autoencoder. However, this does not address the problem of blurriness exposed in the previous section, which is why we opt for a loss structure that is similar to LPIPS, where the loss term is defined over the intermediate feature spaces of the decoder.

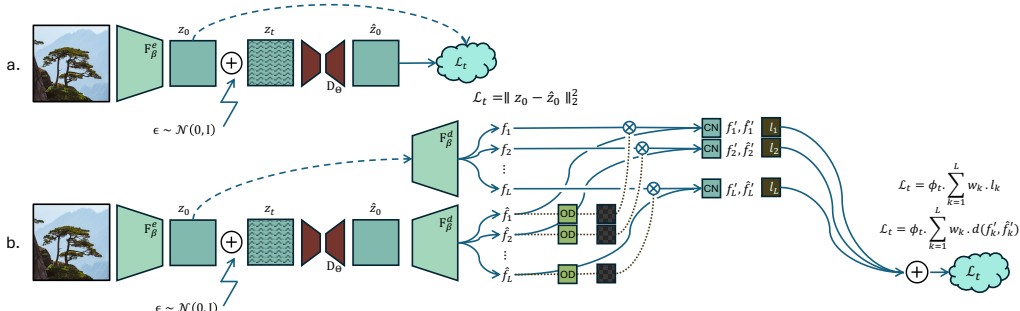

Figure 2: **Overview of our approach.** (a) Latent diffusion models compare clean latents and the predicted latents. (b) Our LPL acts in the features of the autoencoder's decoder effectively aligning the diffusion process with the decoder. $F_\beta^e, F_\beta^d$: *autoencoder encoder and decoder*, $D_\Theta$: *denoiser network*, *CN: cross normalization layer, OD: outlier detection.*

| Framework | $\alpha_t$ | $\sigma_t$ | $\hat{\mathbf{x}}_0$ |
|---|---|---|---|
| DDPM-$\epsilon_t$ | $\sqrt{\bar{\alpha}_t}$ | $\sqrt{1-\bar{\alpha}_t}$ | $(\mathbf{x}_t - \sigma_t D(\mathbf{x}_t, t; \Theta))/\alpha_t$ |
| DDPM-$v_\theta$ | $\sqrt{\bar{\alpha}_t}$ | $\sqrt{1-\bar{\alpha}_t}$ | $\alpha_t \mathbf{x}_t - \sigma_t D(\mathbf{x}_t, t; \Theta)$ |
| Flow-OT | $1-t$ | $t$ | $\mathbf{x}_t - \sigma_t D(\mathbf{x}_t, t; \Theta)$ |

Table 1: Summary of the formula for the estimate of the clean image corresponding to the different formulations. Using the following parameterization, $\forall t, \mathbf{x}_t = \alpha_t \mathbf{x}_0 + \sigma_t \boldsymbol{\epsilon}_t$.

We compute two sets of hierarchies of $L$ decoder features, $\{\phi_l\}_{l=1}^L$, and, $\{\hat{\phi}_l\}_{l=1}^L$, by decoding both the original, $\mathbf{z}_0$, and estimated latents, $\hat{\mathbf{z}}_0$ (where for brevity we drop the dependence of $\hat{\mathbf{z}}_0$ on $t$):

$$\begin{cases} \phi_1, ..., \phi_L = \left(F_\beta^{d,l}(\mathbf{z}_0)\right)_{l\in[\![1,L]\!]}, \\ \hat{\phi}_1, ..., \hat{\phi}_L = \left(F_\beta^{d,l}(\hat{\mathbf{z}}_0)\right)_{l\in[\![1,L]\!]}. \end{cases} \qquad (2)$$

Using these intermediate features, we can define our training objective. Our LPL, $\mathcal{L}_{LPL}$, is a weighted sum of the quadratic distances between the feature representations at the different decoding scales, obtained after normalization:

$$\mathcal{L}_{\text{LPL}} = \mathbb{E}_{t\in\mathcal{T}, \boldsymbol{\epsilon}\sim\mathcal{N}(0,I), \mathbf{x}_0\in D_\mathcal{X}} \left[ \delta_{\sigma_t\leq\tau_\sigma} \sum_{l=1}^L \frac{\omega_l}{C_l} \sum_{c=1}^{C_l} \left\| \rho_{l,c}(\hat{\phi}_{l,c}) \odot \left(\phi'_{l,c} - \hat{\phi}'_{l,c}\right) \right\|_2^2 \right]. \qquad (3)$$

where $\phi'_l$ is the standardized version of $\phi_l$ across the channel dimension, $\rho_{l,c}(\hat{\phi}_{l,c})$ is a binary map masking the detected outliers in the feature map $\hat{\phi}_{l,c}$, $\omega_l$ is a depth-specific weighting and $C_l$ the channel dimensionality of the feature tensor. Note that we better explain these terms later in this section. Moreover, to reduce both LPL computational complexity and memory overhead, we only apply our loss for high signal-to-noise ratios (SNR). In particular, we impose a hard threshold $\tau_\sigma$ and only apply the loss if the SNR is higher than it $\delta_{\sigma_t\leq\tau_\sigma}(\sigma_t)$. The LPL loss is applied in conjunction with the standard diffusion loss, resulting in the training objective $\mathcal{L}_{\text{tot}} = \mathcal{L}_{\text{Diff}} + w_{\text{LPL}} \cdot \mathcal{L}_{\text{LPL}}$.

**Depth-specific weighting.** Empirically, we find the loss amplitude at different decoder layers to differ significantly – it grows with a factor of two when considering layers with a factor two increase resolution. To balance the contributions from different decoder layers, we therefore weight them by the inverse of the upscaling factor *w.r.t.* the first layer, *i.e.* $\omega_l = 2^{-r_l/r_1}$ where $r_l$ is the resolution of the $l$-th layer.

**Normalization.** Since the features in the decoder can have significantly varying statistics from each other, we follow Zhang et al. (2018) and normalize them per channel so that the features in every channel in every layer are zero mean and have unit variance. However, normalizing the feature maps corresponding to the original and denoised latents with different statistics can induce nonzero gradients even when the absolute value has been correctly predicted. To obtain a coherent normalization, we use the feature statistics from the denoised latents to normalize both tensors.

### 3.3 GENERALIZATION FOR LATENT GENERATIVE MODELING

While the bulk of our experiments have been conducted on models trained under DDPM (Ho et al., 2020) for noise prediction, we can generalize our method to different frameworks such as diffusion with velocity prediction (Salimans & Ho, 2022) and flow matching (Lipman et al., 2023). To do this, the only requirement is to be able to estimate the original latents from the model predictions. Under general frameworks such as DDPM and flows, we can write the forward equation in the form $\forall t, x_t = \alpha_t \mathbf{x}_0 + \sigma_t \boldsymbol{\epsilon}_t$, where $\alpha_t$ and $\sigma_t$ are increasing (resp. decreasing) functions of $t$. In Table 1, we provide a summary for these different formulations.

## 4 EXPERIMENTAL EVALUATION

In this section, we first present our experimental setup, and then go on to present our main results, as well as qualitative results and a number of ablation studies.

### 4.1 EXPERIMENTAL SETUP

**Datasets.** We conduct an extensive evaluation on three datasets of different scales and distributions: ImageNet-1k (Deng et al., 2009), CC12M (Changpinyo et al., 2021), and S320M: a large internal dataset of 320M stock images. We note that for both ImageNet-1k and CC12M, human faces were blurred to avoid training models on identifiable personal data. For both CC12M and S320M, we recaption the images using Florence-2 (Xiao et al., 2024) to obtain captions that more accurately describe the image content. For each of these datasets, we conduct evaluations at both $256{\times}256$ and $512{\times}512$ image resolution.

**Architectures.** All experiments are performed using the Multi-modal DiT architecture from Esser et al. (2024). We downscale the model size to be similar to Pixart-$\alpha$ (Chen et al., 2024) and DiT-XL/2 (Peebles & Xie, 2023), which corresponds to 28 blocks with a hidden size of 1,536, amounting to a total of 796M parameters. For ImageNet-1k models we condition on class labels, while for the other datasets we condition on text prompts. For our main results, we perform our experiments using the asymetric autoencoder from Zhu et al. (2023). For ablation studies, we revert to the lighter autoencoder from SDXL (Podell et al., 2024).

**Training and sampling.** Unless specified otherwise, we follow the DDPM-$\epsilon$ training paradigm (Ho et al., 2020), using the DDIM (Song et al., 2021) algorithm with 50 steps for sampling and a classifier-free guidance scale of $\lambda = 2.0$ (Ho & Salimans, 2021). Following Podell et al. (2024), we use a quadratic scheduler with $\beta_{\text{start}} = 0.00085$ and $\beta_{\text{end}} = 0.012$. For the flow experiments, we use the conditional OT probability path (Lipman et al., 2023) with the mode sampling with heavy tails paradigm from Esser et al. (2024). Under this paradigm, the model is trained for velocity prediction and evaluated using the Euler ODE solver. For all velocity models, zero terminal SNR schedule is enforced following Lin et al. (2024).

Similar to Chen et al. (2024), we pre-train all models at 256 resolution on the dataset of interest for 600k iterations. We then enter a second phase of training, in which we optionally apply our perceptual loss, which lasts for 200k iterations for 256 resolution models and for 120k iterations for models at 512 resolution.

When changing the resolution of the images, the resolution of the latents changes by the same factor, keeping the same noise threshold $\tau_\sigma$ yields inconsistent results across resolutions. To ensure consistent behavior, we follow Esser et al. (2024), and scale the noise threshold similarly to how the noise schedule is scaled in order to keep the same uncertainty per patch. In practice, this amounts to scaling the threshold by the upscaling factor. The kernel sizes for the morphological operations in the outlier detection algorithm are also scaled to cover the same proportion of the image.

**Metrics.** To evaluate our models, we report results in terms of FID (Heusel et al., 2017) to assess image quality and to what extent the generated images match the distribution of training images, and CLIPScore (Hessel et al., 2021) to assess the alignment between the prompt and the generated image for text-conditioned models. In addition, we report distributional metrics precision and recall (Sajjadi et al., 2018) as well as density and coverage (Naeem et al., 2020) to better understand effects on image quality (precision/density) and diversity (recall/coverage). We evaluate metrics with respect to ImageNet-1k and, for models trained on CC12M and S320M, the validation set of CC12M.

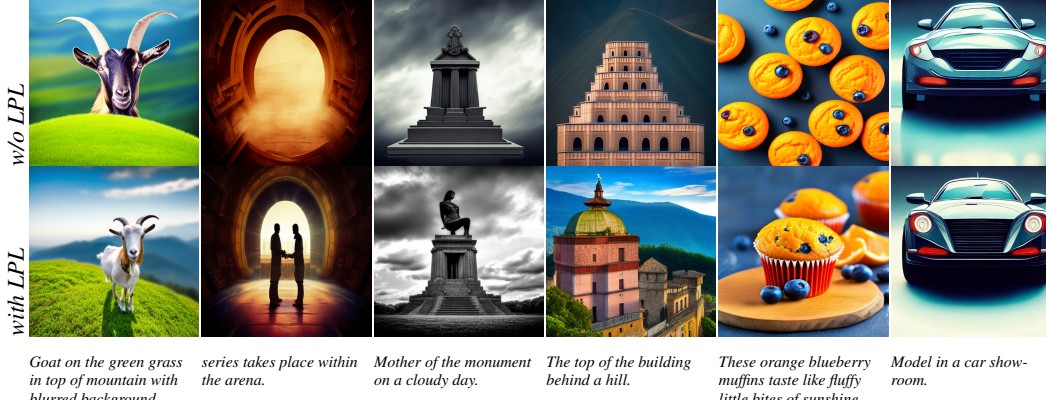

*w/o LPL*    *with LPL*

| Goat on the green grass in top of mountain with blurred background. | series takes place within the arena. | Mother of the monument on a cloudy day. | The top of the building behind a hill. | These orange blueberry muffins taste like fluffy little bites of sunshine. | Model in a car show-room. |

Figure 3: **Samples from models trained with and without our latent perceptual loss on S320M.** Samples from the model with perceptual loss (bottom row) show more realistic textures and details.

| | Pre-training | | Post-training | | | Results | |
|---|---|---|---|---|---|---|---|
| | *Res.* | *Iters* | *Res.* | *Iters* | *LPL* | *FID* ($\downarrow$) | *CLIP* ($\uparrow$) |
| *ImageNet-1k* | 256 | 600k | 256 | 200k | ✗ | 2.98 | — |
| | | | | | ✓ | **2.72** | — |
| | | | 512 | 120k | ✗ | 4.88 | — |
| | | | | | ✓ | **3.79** | — |
| *CC12M* | 256 | 600k | 256 | 200k | ✗ | 7.81 | 25.06 |
| | | | | | ✓ | **6.22** | **25.12** |
| | | | 512 | 120k | ✗ | 8.79 | 24.88 |
| | | | | | ✓ | **7.27** | **25.12** |
| *S320M* | 256 | 600k | 512 | 120k | ✗ | 8.81 | 24.39 |
| | | | | | ✓ | **8.30** | **24.41** |

Table 2: **Impact of our perceptual loss for models trained on different datasets and resolutions for DDPM-$\epsilon$ model**. All models use the same ImageNet-256 pretraining for $600k$ iterations before performing coparing the effect of LPL during post-training. Using LPL boosts FID and CLIP score for all datasets and resolutions considered.

For FID and other distributional metrics, we use the evaluation datasets as the reference datasets and compare an equal number of synthetic samples. For CLIPScore, we use the prompts of the evaluation datasets and the corresponding synthetic samples. Following previous works (Rombach et al., 2022; Peebles & Xie, 2023), we use a guidance scale of 1.5 for resolutions of 256 and 2.0 for resolutions of 512, which we also found to be optimal for our baseline models trained without LPL.

## 4.2 MAIN RESULTS

**LPL applied across different datasets.** In Table 2 we consider the impact of the LPL on the FID and CLIPScore for models trained on the three datasets and two resolutions. We observe that the LPL loss consistently improves both metrics across all three datasets. Most notably, FID is improved by 0.91 points on ImageNet-1k at 512 resolution and by 1.52 points on CC12M at 512 resolution. The CLIPScore is also improved for both resolutions on CC12M, by 0.06 and 0.24 points respectively. Similarly, for the S320M dataset, we observe that FID is improved by 0.51 points while CLIP score improves (marginally) by 0.02 points. Samples of models trained with and without LPL on CC12M and S320M are shown in Figure 1 and Figure 3, respectively.

**Generalization to other frameworks.** We showcase the generality of the LPL by applying it to different generative models, experimenting with DDPM for both epsilon and velocity prediction, and flow matching with optimal-transport (OT) path similar to Esser et al. (2024). In Table 3 we report experimental results for models trained on ImageNet-1k at 512 resolution. The DDPM-based models perform very similar (except perhaps for FID, where they differ by 0.16 points), and we we find significant improvements across all metrics other than density when using LPL. Density remains similar to the baseline for DDPM models, but improves from 1.14 to 1.29 for the Flow-OT model, where all metrics are improved relative to the DDPM trained ones. We posit that this is due to the mode sampling scheme in (Esser et al., 2024), which emphasizes middle timesteps that could better control the trajectories of the flow path towards having more diversity and not improving the quality (precision/density). Hence, applying LPL to Flow-OT solve this by considerably boosting

| Paradigm | DDPM-$\epsilon$ | | DDPM-$v_\Theta$ | | Flow-OT | |
|---|---|---|---|---|---|---|
| LPL | ✗ | ✓ | ✗ | ✓ | ✗ | ✓ |
| FID ($\downarrow$) | 4.88 | **3.79** | 4.72 | **3.84** | 4.54 | **3.61** |
| Coverage ($\uparrow$) | 0.80 | **0.82** | 0.80 | **0.83** | 0.82 | **0.85** |
| Density ($\uparrow$) | **1.14** | 1.13 | **1.15** | 1.14 | 1.14 | **1.29** |
| Precision ($\uparrow$) | 0.74 | **0.77** | 0.73 | **0.78** | 0.75 | **0.79** |
| Recall ($\uparrow$) | 0.49 | **0.51** | 0.49 | **0.50** | 0.52 | **0.54** |

Table 3: **Effect of LPL on ImageNet-1k models at 512 resolution trained with different methods.** We observe consistent improvements on all metrics when incorporating the LPL, except for density metric for which we observe a very slight degradation when using DDPM training.

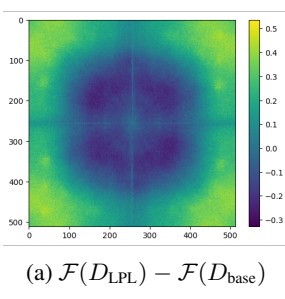
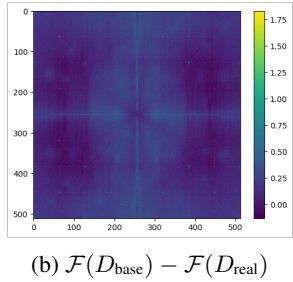
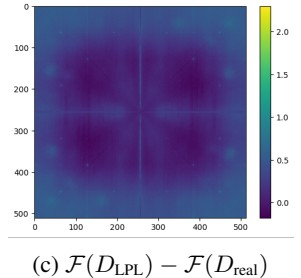

(a) $\mathcal{F}(D_{\text{LPL}}) - \mathcal{F}(D_{\text{base}})$     (b) $\mathcal{F}(D_{\text{base}}) - \mathcal{F}(D_{\text{real}})$     (c) $\mathcal{F}(D_{\text{LPL}}) - \mathcal{F}(D_{\text{real}})$

Figure 4: **Power spectrum of real and generated images.** Difference in (log) power spectrum between image generated with and without LPL. Using LPL strenghtens frequencies at the extremes (very low and very high).

quality. Notably, considering DDPM baselines, LPL provides a boost as significant as the one provided by using flow matching (scores of DDPM w/ LPL in 2nd and 4th columns are on par or better than Flow-OT w/o LPL in 5th column). Moreover, the provided boost is orthogonal to the training paradigm, leading to overall best results when using LPL with the flow model.

**Frequency analysis.** While the metrics above provide useful information on model performance, they do not specifically provide insights in terms of frequencies at which using LPL is more effective at modeling data than the baseline. To provide insight on the effect of our perceptual loss *w.r.t.* the frequency content of the generated images, we compare the power spectrum profile of images generated with a model trained with and without LPL on CC12M at 512 resolution as well as a set of real images from the validation set.

In Figure 4, we plot the difference between log-power spectra between the three image sets. The left-most panel clearly shows the presence of more high frequency signal in the generated images when using LPL to train the model, confirming what has been observed in the qualitative examples of Figure 1 and Figure 3. Moreover, the very lowest frequencies are also strengthened in the samples of the model with LPL. We posit that using the LPL makes it easier to match very low frequencies as they tend be encoded separately in certain channels of the decoder. In Figure 5, we report the error when comparing the power-spectrum of synthetic images and real images, averaged across the validation set of CC12M. For this, we compute the average of the power spectrum across a set of 10k synthetic images from each model and the reference images for the validation set of CC12M. Our experiments indicate that the model trained with LPL is consistently more accurate in modeling high frequencies ($f > 150$), at the expense of a somewhat larger error at middle frequencies ($75 < f < 150$).

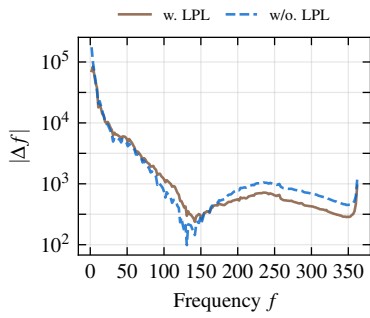

Figure 5: **Frequency comparison.** We compare the power spectrum of the images obtained with or without LPL with real reference images from the validation set of CC12M.

### 4.3 ABLATION RESULTS

**LPL depth.** Using decoder layers to compute our perceptual loss comes with increased computational and memory costs. We therefore study the effect of computing our perceptual loss using only a subset of the decoder layers, as well as a baseline using the RGB pixel output of the decoder. We progressively add more decoder features, so the model with five blocks contains features from the first up to the fifth block. From the results in Figure 7, we find that earlier blocks do not significantly

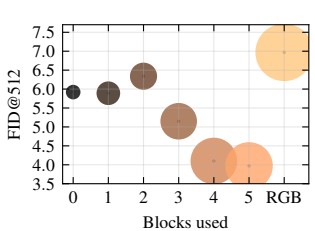

Figure 6: **Ablation study on the impact of the noise threshold $\tau_\sigma$.** We report FID, coverage, density, precision and recall. The dashed line corresponds to the baseline without LPL, note the logarithmic scaling of the noise threshold on the horizontal axis.

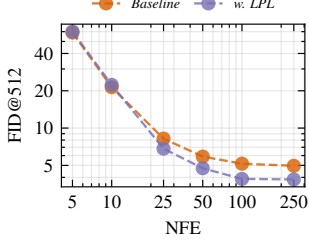

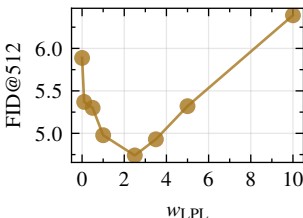

Figure 7: **Exploration of LPL depth.** Influence of decoder blocks used in LPL on FID, 0 corresponds to not using LPL. Disk radius shows GPU memory usage: w/o LPL=64.9 GB, LPL-5 blocks=83.4 GB.

Figure 8: **Impact of LPL for different number of sampling steps.** With higher numbers of sampling steps, the difference between the baseline and the model trained with LPL increases.

Figure 9: **Influence of the LPL loss weight on model performance.** The curve shows a sharp decrease in FID before going back up for larger weights.

improve FID - and can even negatively impact performance. Deeper layers, on the other hand, significantly improve the performance. The most significant gains are obtained when incorporating the third and fourth decoder layers which both improve the FID by more than one point *w.r.t.* the model incorporating one block less. Finally, the last decoder layer improves the FID only marginally and could be omitted to reduce resource consumption. We perform an additional ablation where the loss operates directly on the RGB image space (without using internal decoder features), which results in degraded performance compared to the baseline while inducing a considerable memory overhead.

**Feature normalization.** Before computing our perceptual loss, we normalize the decoder features. We compared normalizing the features of the original latent and the predicted one separately, or normalizing both using the statistics from the predicted latent. Our experiment is conducted on ImageNet-1k at 512 resolution. While the model trained with separately normalized latents results in a slight boost of FID (4.79 *vs.* 4.88 for the baseline w/o LPL), the model trained with shared normalization statistics leads to a much more significant improvement and obtains an FID of 3.79.

**SNR threshold value.** We conduct an experiment on the influence of the SNR threshold which determines at which time steps our perceptual loss is used for training. Lower threshold values correspond to using LPL for fewer iterations that are closest to the noise-free targets. We report results across several metrics in Figure 6 and illustrated with qualitative examples in the supplementary material in Figure 20. We find improved performance over the baseline without LPL for all metrics and that the best values for each metric are obtained for a threshold between three and six, except for the recall which is very stable (and better than the baseline) for all threshold values under 20.

**Reweighting strategy.** We compare the performance when using uniform or depth-specific weights to combine the contributions from different decoder layers in the LPL. We find that using depth-specific weights results in significant improvements in terms of image quality *w.r.t.* using uniform weights. While the depth-specific weights achieve an FID of 3.79, the FID obtained using uniform weights is 4.38. Hence, while both strategies improve image quality over the baseline (which achieves an FID of 4.88), reweighting the layer contributions to be approximately similar further boosts performance and improves FID by 0.59 points.

**LPL and convergence.** As the LPL loss adds a non-negligible memory overhead, by having to backpropagate through the latent decoder, it is interesting to explore at which point in training it should be introduced. We train models on ImageNet-1k at 512 resolution with different durations of the post-training stage. We use an initial post-training phase — of zero, 50k, or 400k iterations — in which LPL is not used, followed by another 120k iterations in which we either apply LPL or not.

| Initial post-train iters | 0 | 50k | 400k |
|---|---|---|---|
| $\Delta$ FID ($\downarrow$) | $-0.58$ | $-0.78$ | $\mathbf{-0.97}$ |
| $\Delta$ coverage ($\uparrow$) | $\mathbf{+4.29}$ | $+3.51$ | $+3.99$ |
| $\Delta$ density ($\uparrow$) | $+0.14$ | $+0.12$ | $\mathbf{+0.21}$ |
| $\Delta$ precision ($\uparrow$) | $+4.01$ | $+4.55$ | $\mathbf{+5.89}$ |
| $\Delta$ recall ($\uparrow$) | $+1.99$ | $+2.32$ | $\mathbf{+4.22}$ |

Table 4: **Effect of post-training with LPL.** In each column, we report the difference in metrics after post-training for 120k iterations with or without LPL. All metrics improve when adding LPL in the post-training phase.

The results in Table 4 indicate that in each case LPL improves all metrics and that the improvements are larger when the model has been trained longer and is closer to convergence (except for the coverage metric where we see the largest improvement when post-training for only 120k iterations). This suggests that better models (ones trained for longer) benefit more from our perceptual loss.

**Influence on sampling efficiency.** We conduct an experiment to assess the influence of the perceptual loss on the sampling efficiency. To this end, we sample the ImageNet@512 model with different numbers of function evaluations (NFE) then check the trends for the baseline and the model trained with our method. For this experiment, we use DDIM algorithm. Results are reported on Figure 8, where we find that for very low numbers of function evaluations, both models perform similarly. The improvement gains from the LPL loss start becoming considerable after 25 NFEs, where we observe a steady increase in performance gains with respect to the number of function evaluations up to 100, afterwards both models stabilize at a point where the model trained using LPL achieves an improvement of approximately 1.1 points over the baseline.

**Impact on EMA.** Since the LPL has the effect of increasing the accuracy of the estimated latent during every timestep, it reduces fluctuations between successive iterations of the model during training. Consequently, when training with LPL the EMA momentum can be reduced to obtain optimal performance. In Figure 10 we report the results of a grid search over the momentum parameter $\gamma_{\text{EMA}}$. We find that the model trained with LPL achieves better results when using a slightly lower momentum than the baseline. From the graph, it's clear that better FID is obtained closer to $\gamma_{\text{EMA}} = 0.99975$ for the LPL model, which corresponds to a half-life of approximately 2750 iterations, while the non-LPL model achieves its optimal score at $\gamma_{\text{EMA}} = 0.9999$ corresponding to a half life of 6930 iterations, more than twice as much as the LPL model, thereby validating our hypothesis.

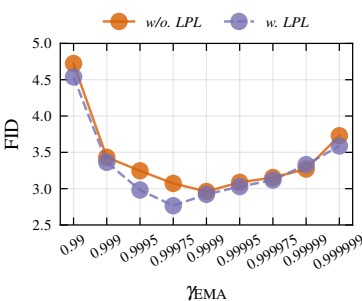

Figure 10: **Impact of EMA decay rate.** Training with LPL is more stable, and allows for a smaller decay parameter.

**Relative weight.** We conduct a grid search over different values for the weight of the LPL loss $w_{\text{LPL}}$. We report FID after training for 120k iterations at 512 resolution, all models are initialized from the same 256 pretrained checkpoint. Our results are reported in Figure 9. Introducing LPL sharply decreases FID for lower weights before going back up at higher weights. We find the model to achieve the best FID for $w_{LPL} \approx 3.0$ which roughly corresponds to a fifth of the relative contribution to the total loss.

## 5 CONCLUSION

In this work, we identified a disconnect between the decoder and the training of latent diffusion models, where the diffusion model loss does not receive any feedback from the decoder resulting in perceptually non-optimal generations that oftentimes lack high frequency details. To alleviate this disconnect we introduced a latent perceptual loss (LPL) that provides perceptual feedback from the autoencoder's decoder when training the generative model. Our quantitative results showed that the LPL is generalizable and improves performance for models trained on a variety of datasets, image resolutions, as well as generative model formulations. We observe that our loss leads to improvements from 6% up to 20% in terms of FID. Our qualitative analysis show that the introduction of LPL leads to models that produce images with better structural consistency and sharper details compared to the baseline training. Given its generality, we hope that our work will play an important role in improving the quality of future latent generative models.

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
