# A APPENDIX

## A.1 RELEVANCE OF THE LPL LOSS

**LPL as an image-space projection penalty.** In the following, we interpret LPL in the simple setting where the decoder is approximated by a linear mapping $F_\beta^d(\mathbf{z}) = \hat{\mathbf{x}} = \mathbf{A}\mathbf{z}$.

Under the DDPM paradigm, for latent diffusion, a neural network is trained to model the reverse process $q(\mathbf{z}_{t-1}|\mathbf{z}_t)$. Under this setting, training is conducted by optimizing the KL divergence between the true reverse process and the predictor that is modeled using a neural network:

$$\mathcal{L}_{t-1} = \mathbb{E}_q \left[ D_{\text{KL}} \left( q(\mathbf{z}_{t-1}|\mathbf{z}_t, \mathbf{z}_0) \parallel p_\Theta(\mathbf{z}_{t-1}|\mathbf{z}_t) \right) \right]. \tag{4}$$

Taking into account the global objective which is image generation, we argue that the accuracy of the sampling steps should be measured in image space and not in latent space, this means *putting more emphasis on obtaining $\ell_2$-optimal reconstructions in image space rather than in latent space*. Such a constraint can be imposed in the form of a penalty term that is added to the training objective:

$$\mathcal{L}_{t-1}^{\text{pen}} = \mathbb{E}_q \left[ D_{\text{KL}} \left( q(\mathbf{x}_{t-1}|\mathbf{x}_t, \mathbf{x}_0) \parallel p_\Theta(\hat{\mathbf{x}}_{t-1}|\hat{\mathbf{x}}_t) \right) \right]. \tag{5}$$

Following DDPM Ho et al. (2020) notation, the ground truth and predicted forward process posterior distributions are given by:

$$q(\mathbf{x}_{t-1}|\mathbf{x}_t, \mathbf{x}_0) = \mathcal{N} \left( \mathbf{x}_{t-1}; \tilde{\boldsymbol{\mu}}_t \left( \mathbf{x}_t, \mathbf{x}_0 \right), \tilde{\beta}_t \mathbf{I} \right) \tag{6}$$

$$p_\Theta(\hat{\mathbf{x}}_{t-1}|\hat{\mathbf{x}}_t) = \mathcal{N} \left( \hat{\mathbf{x}}_{t-1}; \boldsymbol{\mu}_\Theta(x_t, t), \sigma^2 \mathbf{I} \right). \tag{7}$$

In this formula, we can introduce the linear mapping to observe the latent variables instead.

$$q(\mathbf{x}_{t-1}|\mathbf{x}_t, \mathbf{x}_0) = \mathcal{N} \left( \mathbf{x}_{t-1}; \mathbf{A}\tilde{\boldsymbol{\mu}}_t(\mathbf{z_t}, \mathbf{z}_0), \tilde{\beta}\mathbf{A}\mathbf{A}^\top \right) \tag{8}$$

$$p_\Theta(\hat{\mathbf{x}}_{t-1}|\hat{\mathbf{x}}_t) = \mathcal{N} \left( \hat{\mathbf{x}}_{t-1}; \mathbf{A}\boldsymbol{\mu}_\Theta(\mathbf{z}_t, t), \sigma^2 \mathbf{A}\mathbf{A}^\top \right). \tag{9}$$

Subsequently, eq. (5) can be developed using the closed form of KL divergence between two Gaussian distributions:

$$\mathcal{L}_{t-1}^{\text{pen}} = \frac{1}{2}\mathbb{E}_q \left[ \log \frac{|\sigma^2|}{|\tilde{\beta}_t|} + (\mathbf{A}(\tilde{\boldsymbol{\mu}}_t - \boldsymbol{\mu}_\Theta))^\top \sigma^{-2} \left( \mathbf{A}\mathbf{A}^\top \right)^{-1} (\mathbf{A}(\tilde{\boldsymbol{\mu}}_t - \boldsymbol{\mu}_\Theta)) + \left( \frac{\sigma^2}{\tilde{\beta}_t} - 1 \right) \text{Tr} \left\{ \mathbf{I} \right\} \right] \tag{10}$$

$$\mathcal{L}_{t-1}^{\text{pen}} = \frac{1}{2\sigma^2}\mathbb{E}_q \left[ \left( \mathbf{A}(\tilde{\boldsymbol{\mu}}_t - \boldsymbol{\mu}_\Theta) \right)^\top \left( \mathbf{A}\mathbf{A}^\top \right)^{-1} \left( \mathbf{A}(\tilde{\boldsymbol{\mu}}_t - \boldsymbol{\mu}_\Theta) \right) \right] + \text{C}. \tag{11}$$

An upper bound for this term can be obtained by taking the largest eigenvalue of the pseudo-inverse $\left( \mathbf{A}\mathbf{A}^\top \right)^{-1}$.

$$\mathcal{L}_{t-1}^{\text{pen}} \leq \frac{1}{2\sigma^2} \lambda_{\max} \left( \left( \mathbf{A}\mathbf{A}^\top \right)^{-1} \right) \mathbb{E}_q \left[ \|\mathbf{A}(\tilde{\boldsymbol{\mu}}_t - \boldsymbol{\mu}_\Theta)\|_2^2 \right] + \text{C}. \tag{12}$$

From this equation, the image-level penalty can be interpreted as equivalent to optimizing the reconstruction between the real image and the decoded latent prediction $\|\mathbf{A}\hat{\mathbf{z}}_0(\mathbf{z}_t, t; \Theta) - \mathbf{x}_0\|_2^2$.

In the more general case, where the decoder is not a linear mapping, we can use a first order Taylor expansion to obtain a linear approximation, assuming that $\mathbf{z}_0$ and $\hat{\mathbf{z}}_0$ are close enough, which is reasonable for earlier timesteps.

$$\hat{\mathbf{z}}_0(\mathbf{z}_t, t; \Theta) = D(\mathbf{z}_t, t; \Theta) = \mathbf{z}_0 + \sigma_{\mathbf{z}_t}\mathbf{n}, \tag{13}$$

where $\sigma_{\mathbf{z}_t}\mathbf{n}$ is small compared to $\mathbf{z}_0$.

$$\hat{\mathbf{x}}_0(\mathbf{z}_t, t; \Theta) = F_\beta^d(\hat{\mathbf{z}}_0) = \mathbf{x}_0 + \sigma_{\mathbf{z}_t}\nabla_{\mathbf{z}_0}F_\beta^d(\mathbf{z})^\top \mathbf{n}. \tag{14}$$

Which brings us back to the linear case that was tackled above. Consequently, for earlier timesteps, LPL loss can be interpreted as an image space penalty that pushes the forward process posteriors to be more accurate in images space rather than in latent space.

## A.2 LATENT STRUCTURE

Because of the underlying structure of the latent space, certain errors can have much more detrimental effects to the quality of the decoded image than others. We illustrate this in Figure 11 by comparing the generated image after interpolating the encoded latents to different resolutions then back to its original resolution before decoding them. While these different transformations yield similar errors in terms of MSE, especially in RGB space, the interpolation algorithm becomes crucial when working in the latent space.

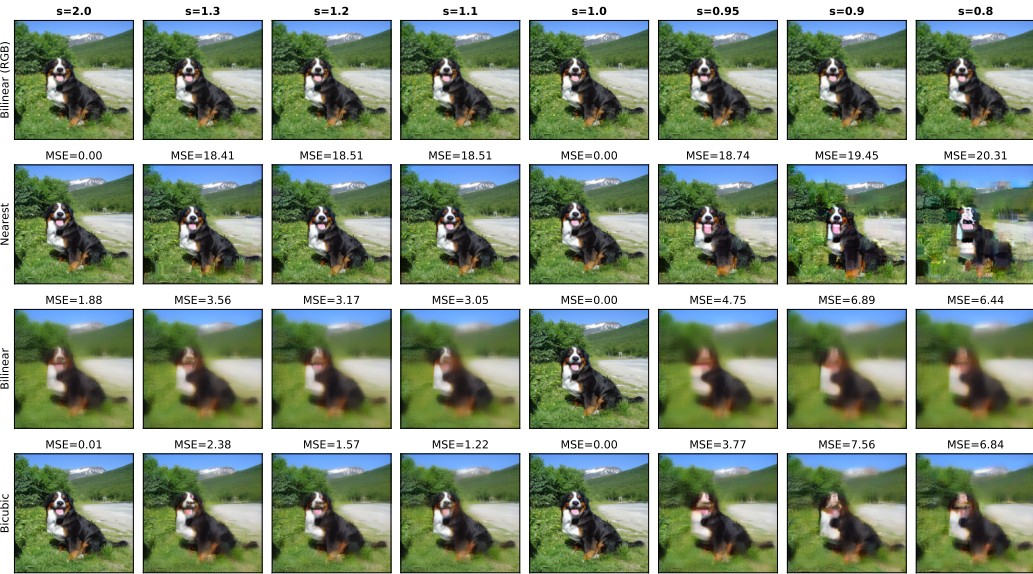

Figure 11: **Influence of interpolation artefacts on latent reconstruction.** We downscale the image by a factor of $1/s$ before upscaling back to recover the original resolution. *From top to bottom:* bilinear interpolation in pixel space, nearest in latent space, bilinear in latent space and bicubic interpolation in latent space.

An illustration of this effect is presented in Figure 11 where we degrade the quality of the latents by performing an interpolation operation to downsize the latents (when $s < 1$) followed by the reverse operation to recover latents at the original size, such a transformation can be seen as a form of lossy compression where different interpolation methods induce different biases in the information lost.

By examining the reconstructions from the latents, we cannot conclude that there is a direct relationship between the MSE with respect to the original latent and the decoded image quality. While nearest interpolation results in the highest MSE, the reconstructed images are more perceptually similar to the target than the ones obtained with bilinear interpolation. Similarly, while the bicubic interpolation with $s = 1.3$ achieves an MSE of 2.38, it still results in better reconstruction than the bilinear interpolation where $s = 2.0$ which achieves a lower MSE error of 1.88.

From this analysis, we see that certain kinds of errors can have more or less detrimental effects on the image generation, which go beyond simple MSE in the latent space.

Another experiment illustrating the irregularity of the autoencoder's latent space is given in Figure 12. In this experiment, we select certain pixel in latent space to which we add a slight random noise (that is half the variance of the latents), we afterwards reconstruct the image from the perturbed latent. What we can see is that depending on the perturbed region, the error in image space can differ significantly. Most notably, we uncover many cases where masking a region in latent space can degrade the reconstruction quality over the whole image. This also showcases that certain high-level information can be contained in certain spatial regions of the latents while others play a relatively less important role.

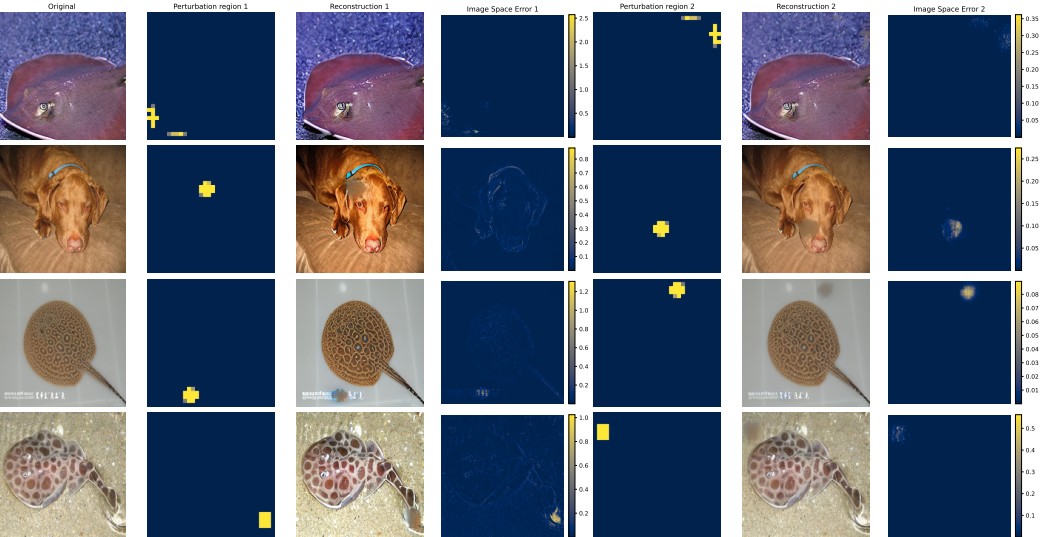

Figure 12: **Illustration of the irregularity of the SD AE.** Certain regions of the image induce a global error in the image and higher error norm, while the same perturbation in other locations in the image results in much lower and localized error.

## A.3 OUTLIER DETECTION

At deeper layers of the autoencoder, some layers have aretfacts where small patches in the feature maps have a norm orders of magnitude higher than the rest of the feature map. These aretfacts have been detected consistently when testing the different opensource autoencoders available online, which include the ones used in our experiments[1], as well as others.[2]

**Outlier detection.** When inspecting the decoder features we find artefacts at decoder's deeper layers. Particularly, in some cases a small number of decoder activations have very high absolute values, see Figure 13. This is undesirable, as such outliers can dominate the perceptual loss, reducing its effectiveness. To prevent this, we use a simple outlier detection algorithm to mask them when computing the perceptual loss. See the supplementary material for details.

To ensure easy adaptability to different models, we propose a simple detection algorithm for these patches and mask them when computing the loss and normalizing the feature maps. Our algorithm is based on simple heuristics and is not meant to provide a state-of-the-art solution for outlier detection. Rather, it is proposed as a temporary patch for the observed issues, while the long-term solution would be to train better autoencoders that do not suffer from these outliers.

**Detection algorithm.** We empirically observe that the activations for every feature map approximately follow a normal distribution, while the outliers can be identified as a small subset of out-of-distribution points. To identify them, we threshold the points with the corresponding percentile at $\delta_o$ and $1 - \delta_o$ percentiles. Since computing the quantiles can be computationally expensive during training, we do it using nearest interpolation, which amounts to finding the $k$-th largest value in every feature map where $k = \delta_o \times H_f \times W_f$ (or $k = (1 - \delta_o) \times H_f \times W_f$ for the maximal values). To remove small false positives that persist in the outlier mask, we apply a morphological opening, which can be seen as an erosion followed by a dilation of the feature map. Pseudo-code for the outlier detection algorithm is provided in Alg. 1.

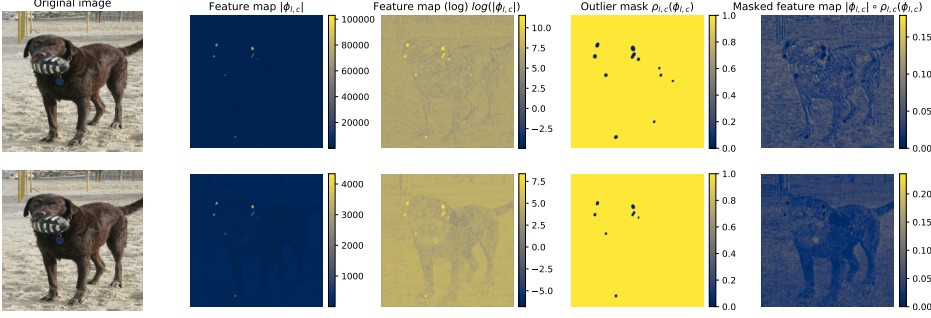

Figure 13: **Example of feature maps from the autoencoder's decoder.** The presence of outliers makes the underlying feature representation difficult to exploit. $l$ refers to the block index, while $c$ is the channel index within the block. Top row: $l = 4, c = 2$, bottom row: $l = 4, c = 8$.

[1]https://huggingface.co/stabilityai/sdxl-vae, and https://huggingface.co/cross-attention/asymmetric-autoencoder-kl-x-1-5
[2]https://huggingface.co/CompVis/stable-diffusion-v1-4, and https://huggingface.co/cross-attention/asymmetric-autoencoder-kl-x-2.

```python
def remove_outliers(features, down_f=1, opening=5,
                    closing=3, m=100, quant=0.02):
    opening = int(ceil(opening/down_f))
    closing = int(ceil(closing/down_f))

    if opening == 2:
        opening = 3
    if closing ==2:
        closing = 1

    # replace quantile with kth (nearest interpolation).
    feat_flat = features.flatten(-2, -1)

    k1 = int(feat_flat.shape[-1]*quant)
    k2 = int(feat_flat.shape[-1]*(1-quant))

    q1 = feat_flat.kthvalue(k1, dim=-1).values[..., None, None]
    q2 = feat_flat.kthvalue(k2, dim=-1).values[..., None, None]

    # Mask obtained by thresholding at the upper quantiles.
    m = 2*feat_flat.std(-1)[..., None, None].detach()
    mask = (q1-m < features)*(features < q2+m)

    # dilate the mask.
    mask=MaxPool2d(
            kernel_size=closing,
            stride=1,
            padding=(closing-1)//2
        )(mask.float()) # closing

    mask=(-MaxPool2d(
            kernel_size=opening,
            stride=1,
            padding=(opening-1)//2
        )(-mask)).bool() # opening

    features = features * mask
    return mask, features
```

Algorithm 1: **Outlier detection algorithm.** The algorithm works by setting a threshold according to the upper $0.02$ quantile of the activations in the feature map. Because the outliers are orders of magnitude away from the rest, we shift the threshold by an offset $m$ that guarantees that only the outliers are thresholded while no activations are masked when no outliers are present. Subsequently, we smooth out the predicted mask using a dilation operation that eliminates small noise in the mask.

## A.4 Perceptual Losses in Diffusion

In this section, we provide a comparison to recent related works that incorporate some form of perceptual objective into diffusion model training.

**E-LatentLPIPS.** Kang et al. (2024), train a classifier in the autoencoder latent space to define an LPIPS metric. While their models are only tested for distilling diffusion models into GANs, we experiment with using them as training losses and compare their performance with LPL. To this end, we only include the $\ell_2$ objective and do not apply any augmentations to the inputs for the loss.

**Self-Perceptual.** Lin & Yang (2024), develop a Self-Perceptual loss where the intermediate features of the denoiser network are used as basis for the perceptual loss that is used during training. this loss is not directly applicable in our use-case as it was developed on a UNet model, which shows a significantly different structure from the state-of-the-art DiT networks. Most notably, the UNet encoder progressively downsamples the feature maps before upsampling them again to their original size, corresponding blocks of the same resolution are linked with skip connections. Hence, the deepest encoder block results in a semantic map description that can be obtained relatively efficiently. On the other hand, for DiT-type models, the successive blocks all share the same resolution, making it unclear as to which block is optimal for this use-case. We note, however, that Lin & Yang (2024) report best performance when using the standard $\ell_2$ loss with classifier-free guidance, *i.e.* better than when using their Self-Perceptual loss. Therefore its practical usefulness remains unclear. To evaluate this method with state-of-the-art DiT-type architectures, we conduct experiments where the loss is computed at different depths of the network and compare the results with our LPL.

**Comparison.** To ensure a fair comparison, we set the weight of the loss such as the variance of the perceptual loss is $0.1$ of the variance of the $\ell_2$ loss when $\ell_2$ loss is also part of the training objective. Results are summarized in Table 5.

In terms of *training throughput* (using the asymmetric auteoncoder at $512$ resolution), we observe a decrease of approximately $42\%$ when training with LPL *vs.* when not training with it, this is significant but is still manageable considering it only applies at later training stages. In comparison, Self-Perceptual with the middle layer applied to MMDiT reduces throughput by $31\%$, such effect should get larger with larger models, making the use of this method prohibitive for very large transformer architecture, especially if used in conjunction with EMA, in which case three different copies of the model should be kept in memory. E-LatentLPIPS on the other hand is more efficient ($20\%$ throughput reductions) as it uses lightweight models that operate at low resolutions.

In terms of *performance*, none of the models that we compare with achieve better FID scores than the baseline. For Self-Perceptual, we observe the highest degradation when using the mid-layer while shallower layers perform the best, although the impact on training performance is negative when compared to the $\ell_2$ baseline. For E-LatentLPIPS, we also observe a degradation of FID by almost 3 points when comparing with the baseline.

| Loss | FID ($\downarrow$) | throughput (img/s) ($\uparrow$) | % Mem/GPU ($\downarrow$) |
|---|---|---|---|
| $\ell_2$ | 4.88 | 16.6 | 42.5 |
| E-LatentLPIPS | 7.71 | 13.3 | 67.5 |
| E-LatentLPIPS$^\dagger$ | 7.19 | 15.8 | 43.5 |
| Self-Perceptual$_8$ | 5.82 | 13.3 | 70.6 |
| Self-Perceptual$_{16}$ | 13.40 | 11.4 | 75.1 |
| Self-Perceptual$_{28}$ | 10.43 | 9.2 | 82.4 |
| LPL | 3.79 | 9.6 | 74.3 |

Table 5: **Comparison with different perceptual losses in latent space.** For Self-Perceptual, we note the depth at which the loss is computed as a subscript. $\dagger$ corresponds to a perceptual loss that is only applied for later timesteps, similar to LPL. We report FID training throughput and memory usage per GPU (in %) for the same batch size. Throughput and memory measurements are taken during the post-training stage, *i.e.* with perceptual losses applied in all iterations.

## A.5 Indirect effects of LPL

One possible explanation for the improved performance is that it comes from a certain timestep-specific reweighting which puts more emphasis on later timesteps, as samples in the LPL timestep range tend to be present both in both the vanilla $\ell_2$ loss and our LPL. To verify this claim, we compare the performance with a model trained with a timestep specific reweighting that equalizes the contributions of the timesteps across the batch, the re-weighting is applied as follows:

$$w(\sigma_t) = 1 + w_{\text{LPL}} \cdot \frac{\sigma^2_{\mathcal{L}_{\text{LPL}}}}{\sigma^2_{\ell_2}} \cdot \delta_{\sigma_t \leq \tau_\sigma}(\sigma_t), \tag{15}$$

where $\sigma^2_{\mathcal{L}_{\text{LPL}}}$ (resp. $\sigma^2_{\ell_2}$) is the mean variance of the LPL loss (resp. the vanilla $\ell_2$ objective). Such a weighting effectively amplifies later timesteps to have the same contribution when using LPL or not using it. Additionally, we compare the standard epsilon weights with state-of-the-art reweighting strategies such as min-SNR (Hang et al., 2023a), results are reported in table 6.

| Re-weighting | FID ($\downarrow$) |
|---|---|
| Baseline | 4.88 |
| Baseline$^\dagger$ | 5.43 |
| min-SNR | 5.04 |
| LPL | 3.79 |

Table 6: **Influence of timestep reweighting strategy.** Compared to different timestep reweighting strategies, LPL finetuning achieves significant improvements while timestep reweightings result in degraded performance compared to the baseline epsilon weighting. $\dagger$ corresponds to the reweighting from eq. (15).

As seen in Table 6, introducing timestep reweighting results in degraded performance compared to the baseline while LPL significantly improves FID. Hence the improvements cannot be attributed to time-specific reweighting.

A.6 MEMORY OVERHEAD

For a more fair and accurate comparisons with the baseline, we conduct additional experiments where we equalize not the number of iterations but the maximum achievable batch size and the training time, in order to obtain a more accurate sense of the applicability of our loss in practice.

**Memory maximization.** Table 7 compares models trained on the same resources for an equal time duration. This experiment is conducted on ImageNet@512, both models are trained on 2 A100 nodes for a duration of 48 hours. For the baseline with a higher batch size, the learning rate is scaled linearly in order to account for this discrepancy. Under this setting, the baseline trains with a batch size/GPU of 25 for 122.5k iterations, achieving an FID of 4.62 while the model with LPL is trains for 82.5k iterations with a batch size/GPU of 16, achieving an FID of 3.84. When comparing FLOPs, we found both runs to have similar FLOPs per iteration. Hence, while the FID gap between the two models is reduced from 1.09 to 0.78, it remains significant.

| Method | Batch Size/GPU | (T)FLOP/it | Training iterations (k) | FID ($\downarrow$) |
|---|---|---|---|---|
| Baseline | 25 | 108.347 | 122.5 | 4.62 |
| with LPL | 16 | 106.581 | 82.5 | **3.84** |

Table 7: **Comparisons for the same time budget.** All models are trained with maxed-out batch size for 48 hours using two A100 nodes per experiment.

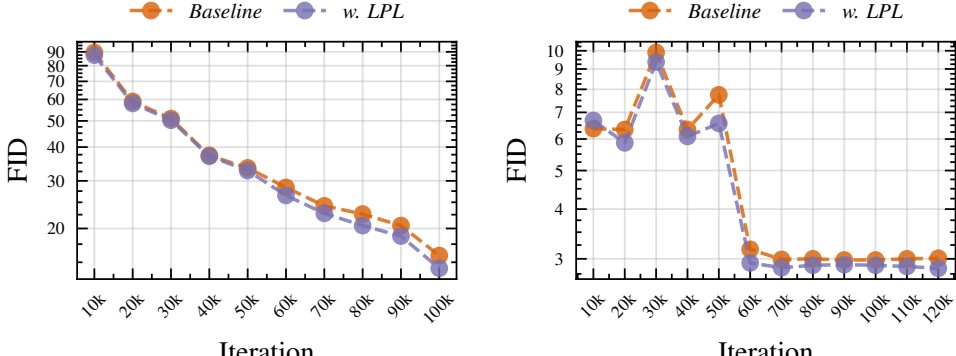

Figure 14: **LPL from scratch.** Comparison on the effect of LPL when applied from scratch.

Figure 15: **LPL finetuning.** Comparison on the effect of LPL when applied as finetuning.

**Minimum iterations.** To quantify the needed number of training iterations, we perform finetuning experiments and track the FID every 10k iterations, allowing us to track how long the LPL should be applied to see benefits. These experiments are conducted on ImageNet-1k at 256 resolution.

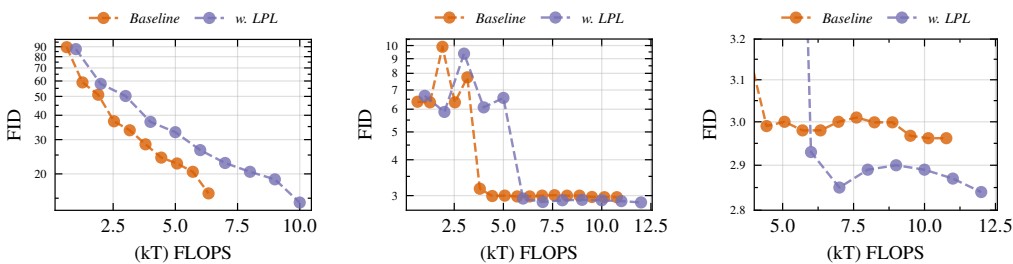

Figure 16: **LPL from scratch.** Comparison on the effect of LPL when applied from scratch.

Figure 17: **LPL finetuning.** Comparison on the effect of LPL when applied as finetuning.

Figure 18: **LPL finetuning.** Crop around EMA region.

| | **Pre-training** | | | **Post-training** | | | | **Total** | | |
|---|---|---|---|---|---|---|---|---|---|---|
| | *Res.* | *Iters* | *(T)FLOPs/it* | *Res.* | *Iters* | *LPL* | *(T)FLOP/it* | *(T)FLOPS* | FLOPs Increase |
| *ImageNet-1k* | 256 | 600k | 63.389 | 256 | 200k | ✗ | 63.342 | $50,711.20$k | 14.43% |
| | | | | | | ✓ | 99.987 | $58,030.80$k | |
| | | | 63.389 | 512 | 120k | ✗ | 69.342 | $46,354.44$k | 9.64% |
| | | | | | | ✓ | 106.581 | $50,823.12$k | |
| *CC12M/S320M* | 256 | 600k | 63.391 | 256 | 200k | ✗ | 63.391 | $50,712.80$k | 14.46% |
| | | | | | | ✓ | 100.064 | $58,047.40$k | |
| | | | 63.391 | 512 | 120k | ✗ | 69.342 | $46,355.64$k | 9.64% |
| | | | | | | ✓ | 106.583 | $50,824.56$k | |

Table 8: **FLOP analysis**. We compare the total FLOP increase throughout training under the different settings in our main results table.

In Figure 16, we experiment with the effect of training from scratch when using LPL. In the beginning of training, both models have similar performance, however as training continues and the performance gain due to LPL increases.

In Figure 18, we experiment with finetuning a pre-trained model and track FID every $10k$ iterations. The big FID improvement at $50k$ iterations is due to the activation of EMA. We observe that as of $20k$ iterations, the model with LPL consistently obtains better performance.

**FLOPS Analysis.** In Table 8, we conduct a detailed analysis of the FLOP count for the duration of training under different settings, reporting the total FLOP count for different training settings. We observe that adding LPL to the training losses increase the FLOPs/iteration by a faactor of approximately $1.5$. However, when accounting for the number of training iterations, we found the total FLOP increase to be around $10 - 15\%$.

A.7 COMPARISON OF BASELINE TO THE STATE OF THE ART

Our baseline setting uses the DDPM paradigm with the state-of-the-art MMDiT architecture from Esser et al. (2024). For the Flow-OT model, we follow the same setting as SD3. To verify that this baseline is competitive with state-of-the-art models, we provide quantitative comparison of our baseline with other open-source models trained on ImageNet at 512 resolution. Accordingly, we follow the same procedure for the models and report FID for the training set of ImageNet using 250 sampling steps for each sample. Our flow model is sampled using an ODE sampler based on the RK4 solver from Chen (2018). The results in Table 9 show that our flow baseline with RK4 solver achieves state-of-the-art results compared to the results reported in the literature.

| Model | FID ($\downarrow$) |
|---|---|
| UNet Rombach et al. (2022) | 4.81 |
| DiT-XL/2 Peebles & Xie (2023) | 3.04 |
| mDT-v2 Gao et al. (2023) | 3.75 |
| SiT-XL/2 (Flow-SDE) Ma et al. (2024) | 2.62 |
| mmDiT-XL/2 (DDPM - DDIM) | 3.02 |
| mmDiT-XL/2 (Flow-OT - ODE) | **2.49** |

Table 9: **Comparison with other baselines.** We compare our baseline training with other models in the literature, our baseline training results in state-of-the art performance. FID scores for each model are taken from their respective papers.

**Guidance scale.** In the Figure 19, we report the FID of the baseline model trained on ImageNet@512 for different guidance scale values. We find the best FID to be achieved for $w_{CFG} = 2.0$, which is the value chosen in our experiments.

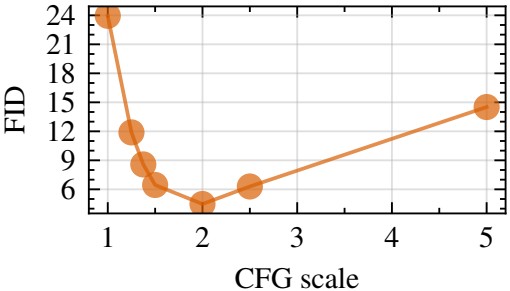

Figure 19: **Influence of guidance scale on FID.**

## A.8 ADDITIONAL QUALITATIVE RESULTS

**Noise threshold.** In Figure 20, we illustrate the impact of using a higher noise threshold (which amounts to using LPL for longer in the diffusion chain) on the image quality. A higher noise threshold yields better structures in the images and exacerbates semantic features that distinguish objects.

**Vanilla diffusion.** In Figure 21, we qualitatively investigate the influence of LPL on a baseline model, without classifier-free guidance and without EMA. We can see that LPL significantly improves the structure of objects as compared to the model that was trained without it.

**Samples of ImageNet-1k models.** In Figure 22 we show samples of models trained with or without LPL on ImageNet-1k at 512 resolution. At higher resolutions, we also observe that the model trained with LPL generates images that are sharper and present more fine-grained details compared to the baseline.

**Samples on T2I models.** We provide additional qualitative comparisons regarding our LPL loss. Figure 23 showcases results on a model trained on CC12M at 512 resolution, Figure 24 showcases results on a model trained on S320M at 256 resolution.

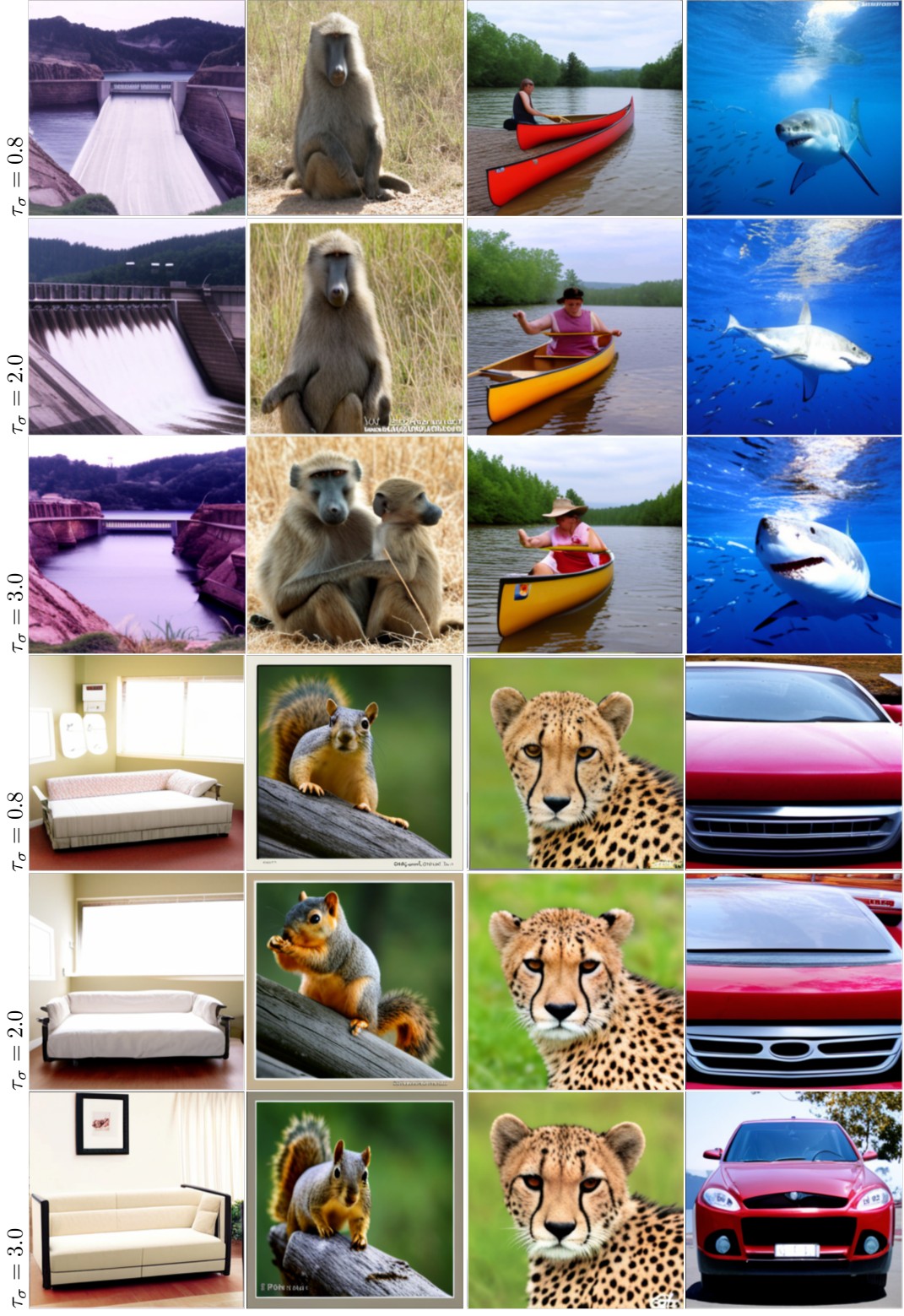

Figure 20: **Influence of noise threshold.** Higher thresholds allow for more detailed and coherent images. Samples obtained from a model trained on ImageNet@256.

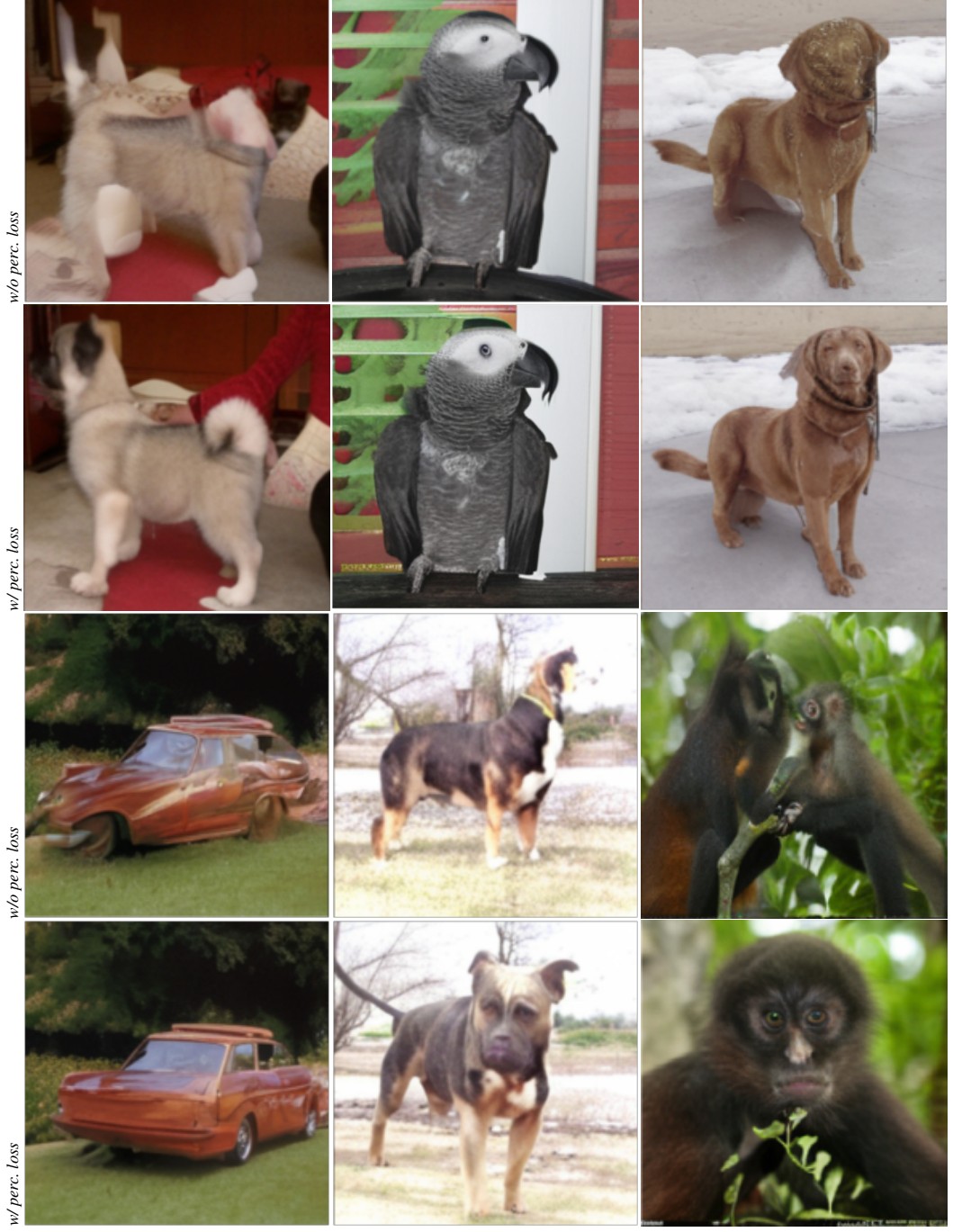

Figure 21: **Qualitative comparison of the effect of the latent peceptual loss.** Models trained on ImageNet-1k at 256 resolution with (bottom) and without (top) our perceptual loss. Without the perceptual loss, the model frequently fails to generate coherent structures, using the perceptual loss, the model generates more plausible objects with sharper details. The models are finteuned for 100k iterations from a checkpoint that was trained for 200k iterations. The samples are generated *without classifier-free guidance or EMA*, using 50 DDIM steps.

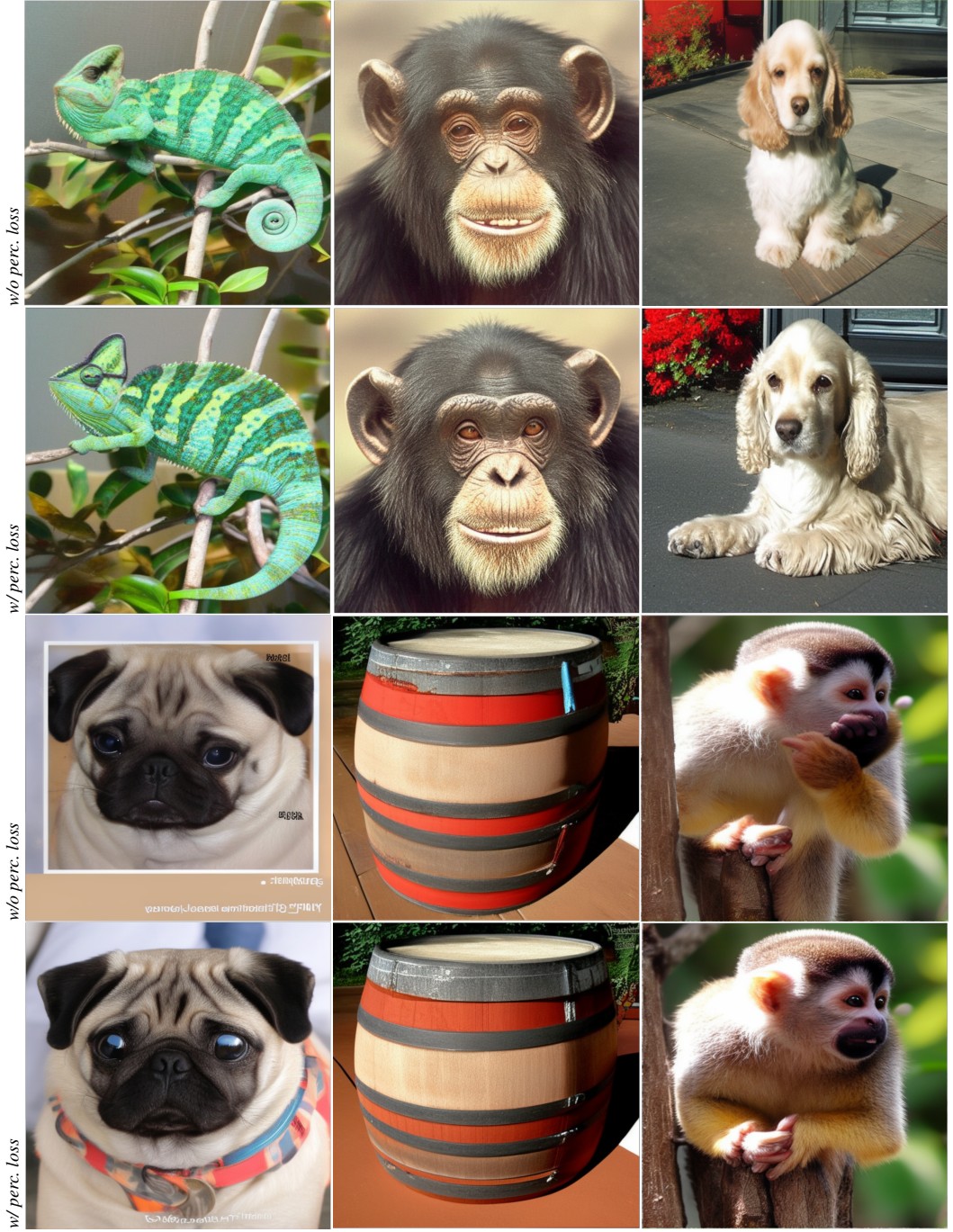

Figure 22: **Influence of finetuning a class-conditional model of ImageNet-1k at 512 resolution using our perceptual loss.** Our perceptual loss (bottom row) leads to more realistic textures and more detailed images.

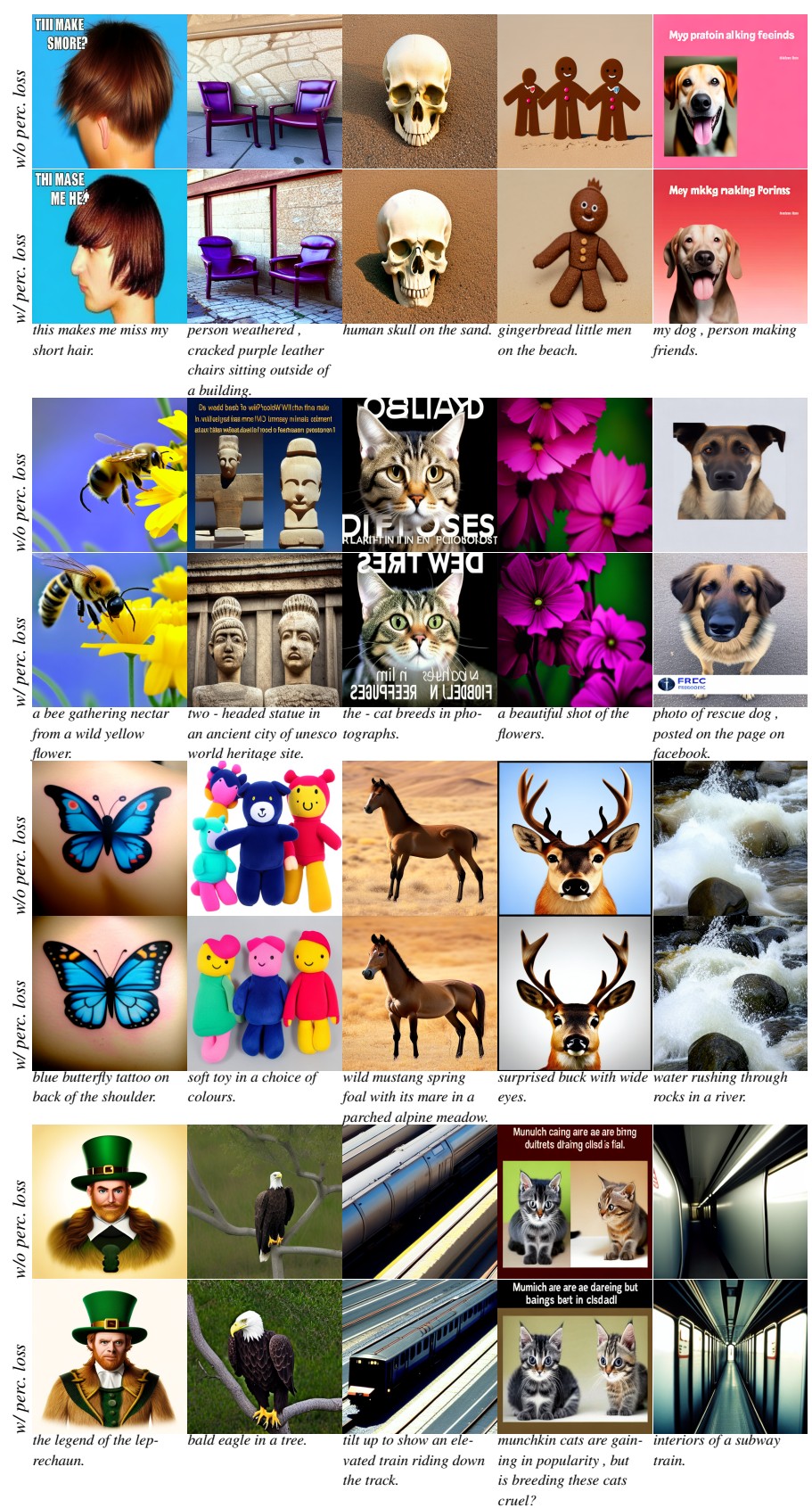

Figure 23: **Qualitative comparison.** Comparison of samples from models trained with and without our perceptual loss on CC12M at 512 resolution (The differences are better viewed by zooming in).

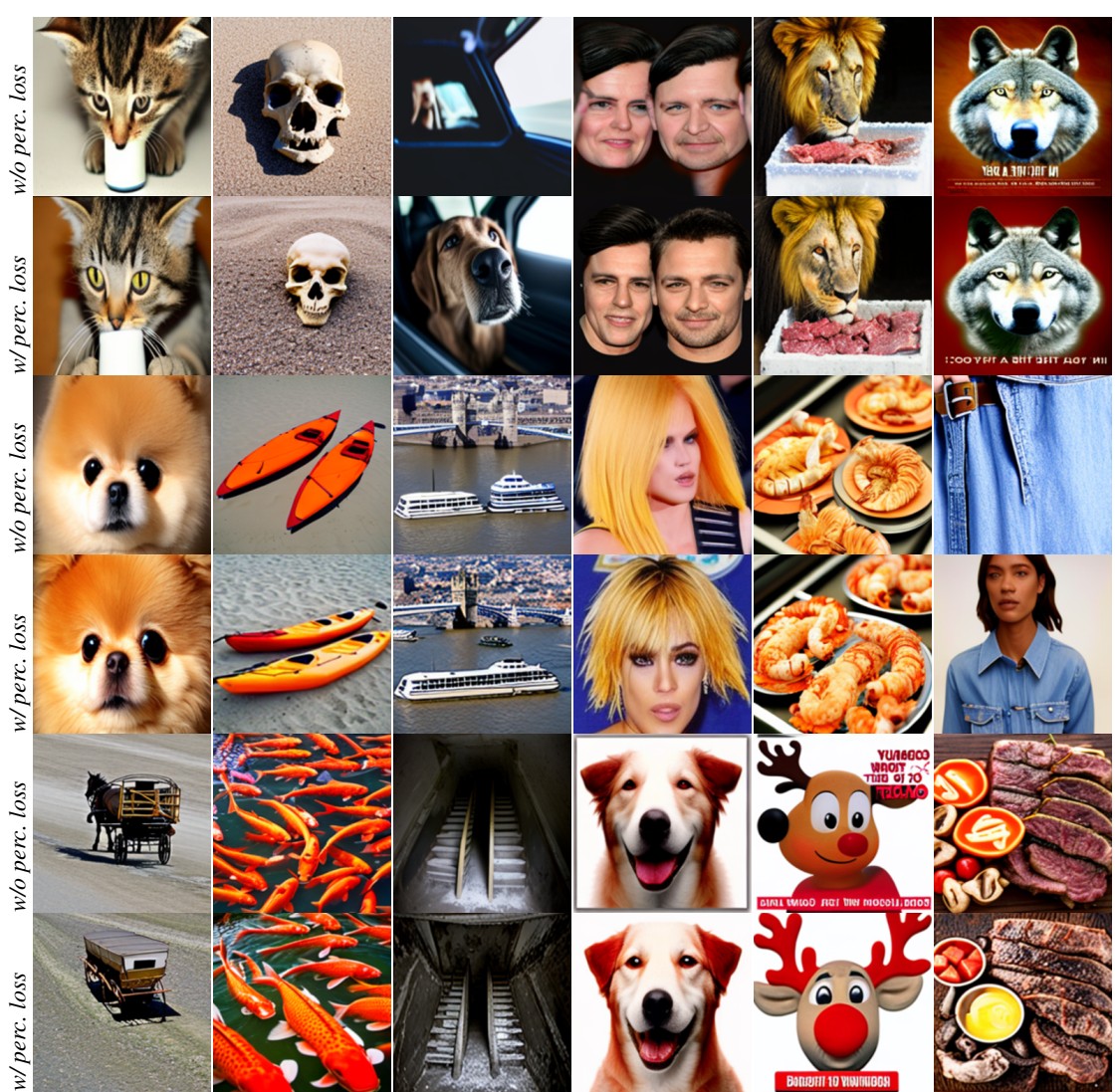

Figure 24: **Qualitative comparison.** of samples from models trained with and without our LPL on S320M at 256 resolution.