# OpenReview forum: "Boosting Latent Diffusion with Perceptual Objectives"
_ICLR.cc/2025/Conference — ICLR 2025 Poster_

### Official Review · Reviewer_W3EV · 2024-10-27

**Soundness:** 3
**Presentation:** 3
**Contribution:** 2
**Rating:** 6
**Confidence:** 3

**Summary:**

This work introduces "latent perceptual loss" (LPL) to improve the training of diffusion models. According to the paper, LPL enhances diffusion models performance across various datasets (Table 2) and methods (Table 3). Additionally, extensive analyses, e.g., comparing the generation quality with and without LPL loss, and ablation studies, deepen our understanding of LPL.

**Strengths:**

**1. Extensive analysis**

This work provides extensive analysis to demonstrate the validity of LPL. For instance, the major performance gain from LPL arises from generating more precise high-frequency components, albeit at the expense of some low-frequency details.

**2. Ablation study**

It gives us various ablation study to provide the motivation behind design choices. The ablation study includes various components including feature depth for LPL, SNR threshold value, Reweighting strategy, or etc. This level of extensive ablation study is rare.

**Weaknesses:**

**1. Efficiency**

In the paper, it is mentioned several times that utilizing the VAE decoder features to compute the LPL loss is costly. This is why recent research has explored alternative perceptual losses, such as latent LPIPS [1]. While the paper claims that LPL incurs minimal computation since it’s only used post-training, this claim is far from acceptable, as the post-training phase in this paper involves iterations amounting to as much as one-third, or at minimum one-fifth, of the pretraining iterations.

[1]: Distilling Diffusion Models into Conditional GANs ([ECCV24](https://mingukkang.github.io/Diffusion2GAN/))

**2. Novelty**

Although subjective, I believe the novelty of LPL is somewhat lacking. This loss trick slightly enhances quality during training but seems more heuristic than principled. Additionally, it doesn’t contribute much new knowledge about diffusion models, which weaken the paper's novelty. Furthermore, it does not appear that any non-trivial trick was devised in the process of introducing the perceptual loss to the diffusion models.

**Questions:**

**1. Efficiency**

- How does the result compare when using FLOPS as the x-axis? Comparing by iteration seems somewhat unfair. If the performance improvement looks promising when compared in terms of FLOPS, I'm considering raising the score.

**2. Novelty**

- Is there a specific reason why LPL was only used in post-training? I feel that LPL might provide a good gradient signal early in training, but it seems that experiment wasn't conducted, which is unfortunate. I'm curious about the performance changes if LPL is applied from the start of the pretraining.

---

> ### Author Response · Authors · 2024-11-21
>
> ### Response W3EV
>
> We thank the reviewer for their encouraging comments: “extensive analysis to demonstrate the validity of LPL”, “This level of extensive ablation study is rare.” Below we address the points raised in the review individually.
>
> **W1 “Efficiency” and relation to Kang et al., ECCV’24.**
> Thanks for bringing this paper of Kang et al. \[1\] under our attention. We consider it concurrent to our work as the ICLR deadline coincided with the ECCV conference.
>
> We discuss the efficiency aspects and computational overhead in the new Section A.5 of the supplementary material.
>
> * Figure 15 shows that LPL can boost performance after as little as 20k iterations when finetuning.
> * Additionally, we show that the gains remain significant even when comparing with a baseline trained with a maxed-out batch size for the same time period.
>
> We discuss the relation with Kang et al in the new Appendix A.4 and compare with their approach experimentally, results are reported in Table 5\. We explored training all the timesteps and only earlier timesteps, similar to our work. In both cases they obtain worse performance than the L2 baseline.
>
> Below, we summarize the main difference between our LPL and E-LatentLPIPS:
>
> * E-LatentLPIPS was only tested for distilling diffusion models into GANs and not for training diffusion models.
> * E-LatentLPIPS requires training of a classifier latent space for every new autoencoder, while LPL uses the already available autoencoder decoder.
> * In terms of interpretation, E-LatentLPIPS does not tackle the mismatch between pixel-space and latent-space diffusion but provides an LPIPS model that can help in latent distillation tasks. As this latent representation is more compressed than the latents, it is unlikely that it will help with high-frequency information similarly to ours.
> * When tested for diffusion training, we obtain results that are degraded with respect to the L2 baseline.
> For more detail we refer to Section A.4 of the supplementary material.
>
> **W3 “Novelty”**
> Our work tackles the mismatch between pixel diffusion and latent diffusion, hence we believe it provides valuable insight about the training dynamics of generative models in latent space, as the latent structure is often an overlooked component in the literature. From our point of view, the novelty comes from identifying the issue of latent-pixel structure mismatch and how it results in suboptimal results.
> We provide extensive experimental results and ablations, showcasing its effectiveness in tackling the identified problem as well as improving overall performance on multiple datasets, resolutions, and training paradigms. We believe this is a valuable contribution to the field.
> In Section A.2 of the supplementary material we provide evidence highlighting the mismatch issue with the current SOTA latent image generative models. Nevertheless, to further strengthen our contributions, we provide an updated version of Section A.1 in the appendix to showcase that LPL can be interpreted as an image-space penalty on the forward process posterior distribution.
> Furthermore, we added a new Section A.4 to the supplementary material with comparisons to other recent works that develop perceptual losses in latent space, showing that just plugging other perceptual losses in the training hurts performance while our approach is more general and improves over the latest SOTA baselines.
>
> **Q1 “How does the result compare when using FLOPS as the x-axis?”**
> We added an extensive analysis on efficiency and computational overhead in the new Section A.6 of the supplementary material.
>
> * Figure 15 shows that LPL can boost performance after as little as 20k iterations when finetuning.
> * Additionally, we show in Table 7 that the gains remain significant even when comparing with a baseline trained with a maxed-out batch size for the same time period.
>
>
> **Q2 “Is there a specific reason why LPL was only used in post-training?”**
> The reason LPL was used in post-training is two-fold:
>
> * Reducing computational overhead.
> * We hypothesize that LPL provides a better signal when denoised predictions are more accurate (this is reflected in Section A.1 of the supplementary material where the condition for the Taylor expansion becomes reasonable), which is also the reason why we impose a threshold on the timesteps for which LPL is applied.
>
> In the new Section A.6 of the appendix, we perform the requested experiments.
>
> * When training from scratch, the model using LPL converges faster than the baseline (Figure 14).
> * When finetuning with LPL, the adaptation period is of less than 20k iterations (Figure 15).
>
> \[1\] Kang et al., Distilling Diffusion Models into Conditional GANs. In ECCV 2024

---

> ### Comment · Reviewer_W3EV · 2024-11-24
>
> Thank you for your dedicated effort on the rebuttal. I really appreciate it.
>
> ---
>
> There were two main issues I pointed out in my review.
>
> **1. Efficiency**
>
> > How does the result compare when using FLOPS as the x-axis? Comparing by iteration seems somewhat unfair.
>
> The authors responded to this in the rebuttal with two strategies:
>
> 1. Even with only 20k iterations in post-training, they already surpass in performance. (Figure 15)
> 2. They outperform when measured using the same GPU and wall-clock time. (Table 7)
>
> Firstly, it's unfortunate that although many reviewers requested comparisons based on FLOPS, the authors did not provide exact FLOPS-based comparisons but instead used iteration (Figure 15). Unfortunately, when comparing based on FLOPS, LPL seems to perform worse (since LPL requires additional x2 more FLOPS). Given this, the technical contribution appears somewhat reduced. If the authors think that comparing LPL based on FLOPS is particularly unfair or disadvantageous, please let me know. If so, I would like to consider that in my evaluation.
>
> Regarding the Table 7, I wonder if the learning rate in the experiment without LPL was appropriately tuned according to the batch size. If so, it helps to convince that the LPL loss helps improve performance under realistic constraints.
>
> **2. Novelty**
>
> > I believe the novelty of LPL is somewhat lacking.
>
> The authors claim that their paper "provides valuable insight about the training dynamics of generative models in latent space." Indeed, as they assert, the role of latent space in latent diffusion models (LDMs) is a relatively less explored area.
>
> However, I am not sure what effective insights the use of LPL provides about the training dynamics of LDMs. From a reader's perspective, the LPL loss is fundamentally a perceptual loss, as indicated by its name, and it's hard to perceive that it solves a fundamental problem between latent space and pixel space. For instance, as other reviewers besides myself have mentioned, they ask whether there are experiments using latent-LPIPS. The underlying meaning of this question is that, in the current flow of the paper, the motivation for using perceptual loss using the VAE decoder is due to perceptual representation, not because of a latent space–pixel space mismatch. **Therefore, if this paper wants to assert novelty, it needs to reorganize the storyline focusing on the mismatch between latent space and pixel space**.
>
> Below are my thoughts on the experiments mentioned in the rebuttal.
>
> > **Section A.1**
>
> I think it provides really good justification. Perhaps this theory could be used to support that it prevents mismatch.
>
> > **Section A.2**
>
> I believe this experiment could serve as a good early demonstration in the paper showing the problem with LDMs. It intuitively shows that while the L2 norm may be appropriate in pixel space, it is not in latent space. It would be beneficial to provide this as initial motivation.
>
> > **Section A.4**
>
> Table 5 shows that LPL is promising compared to the baseline. It's a good result.
>
> > *We provide extensive experimental results and ablations, showcasing its effectiveness in tackling the identified problem as well as improving overall performance on multiple datasets, resolutions, and training paradigms. We believe this is a valuable contribution to the field.*
>
> I indeed agree that this paper brings some improvements in performance. Therefore, even if this paper is accepted, I would not object. And I believe that the contribution of performance-oriented papers is very important. However, I feel that this paper relies heavily on existing methods, w/o giving new insights, at least in current form.
>
> ---
>
> In conclusion, after the rebuttal, the two issues I had—efficiency and novelty—still seem unresolved. It's unclear whether the proposed LPL shows better performance than the baseline w.r.t. FLOPS, and the novel insight that LPL resolves the mismatch between image space and latent space does not seem appropriately conveyed in its current form. I think it would be better to focus less on perceptual loss and discuss the mismatch more thoroughly.
>
> It's painful to say this after you've put so much effort into the rebuttal. If there's any part you want to refute or if I'm misunderstanding and you need to persuade me, please let me know.

---

> ### Author Response · Authors · 2024-11-26
>
> We thank the reviewer for their valuable feedback.
>
> **1\. Efficiency.**
>
> We updated the appendix with FLOPs-based comparisons.
>
> **Table 8** shows the total FLOPs during a training iteration of the different phases present in Table 2\. We find the FLOP count to increase roughly by a factor 1.5 when training with LPL across all configurations. However, when comparing the complete training runs, i.e. taking into account also the (long) pre-training without LPL, we find that the total FLOP count increases by less than 15% (256 resolution models) or less than 10% (512 resolution models), while achieving better FID (shown in Table 2).
>
> **Table 7** now shows the FLOPs/iteration for the training with and w/o LPL, where the batch size has been scaled in both cases to fill the GPU memory. We find that in this case the FLOP count to be actually  slightly higher (1.6%) for the baseline model without LPL but with larger batch size.
>
> **Regarding the learning rate in Table 7**, we did scale the baseline’s learning rate linearly with the batch size (the LPL model uses a learning rate of 10^(-4) while the baseline uses a learning rate of 1.6\*10^(-4). We now explicitly state this in the manuscript.
>
> **Figures 16,17 and 18** show the performance in terms of FID when adopting FLOPs in the x-axis for the baseline model without LPL and the same model with LPL included. In this case the tendency when training from scratch (Fig 17\) is in favour of the baseline which converges faster for similar amounts of FLOPs.  For finetuning (Fig 18), however, we can see that LPL helps the model converge faster and achieves better performance, especially when incorporating EMA, we see that the LPL model converges to a lower point than the baseline. We updated Figures 17/18 with a longer runtime for the baseline in order to equalize the FLOP count.
>
> We hope that these clarifications adequately address the reviewer’s concerns.
>
> **2\. Novelty.**
>
> We thank the reviewer for their suggestion. We follow the reviewer’s advice and reorganize the storyline in the main manuscript in order to focus more on the motivation for our loss (L2 blurriness, pixel-latent mismatch, decoder as perceptual network), while deferring certain implementation details to the appendix (such as the outlier detection approach), we additionally include comparisons with other perceptual losses for diffusion in the main manuscript.
>
> We will continue improving the storyline and the flow of the paper accordingly.

---

> > ### Comment · Reviewer_W3EV · 2024-12-01
> >
> > Here's the English translation of the paper review:
> >
> > > 1. Efficiency
> > I now understand that the learning rate was adjusted in Table 7. That makes sense.
> >
> > I appreciate Figures 16 and 17. When comparing based on FLOPS, the advantages and disadvantages of this method become clearer. I have a question though - it seems that the LPL loss works effectively during the finetuning phase. Do you have any intuition about why LPL might be relatively less effective during the pretraining stage? Recently, a method like [1] showed that image signals only have positive effects in early pretraining, and I was interpreting LPL somewhat from this perspective. However, looking at Figures 16 and 17, this method seems to show a slightly different pattern, being more effective in finetuning, hence my question. By the way, this is just for discussion, and I will not consider it for evaluation.
> >
> > [1]: Representation Alignment for Generation: Training Diffusion Transformers Is Easier Than You Think, https://arxiv.org/abs/2410.06940
> >
> > 2. Novelty
> >
> > As it stands now, there are many aspects that could be improved. It would be better if the writing emphasized LDM's weaknesses, but I trust the authors will work well on this until the camera-ready version.
> >
> > ---
> >
> > Additionally, regarding my previous response, I think I somewhat undervalued the good performance compared to the baselines (Self-perceptual, Latent-LPIPS). This can definitely be considered a performance improvement compared to methods proposed in existing research.
> >
> > In conclusion, some of my concerns have been resolved, and I've adjusted my score accordingly. Thank you for the hard work.

---

> ### Author Response · Authors · 2024-12-04
>
> We thank the reviewer for their thoughtful feedback.
>
> **1 -** One straightforward way to look at this is through the development in A.1.
> There we show that for the image-space divergence penalty to be approximated by our LPL loss, the Taylor expansion rule (l800-809) requires that the predicted denoised latent and the target latent should be similar, such a condition holds better later on during training when the denoiser becomes more accurate.
>
> Broadly speaking, the effect of LPL is a polishing effect on the predicted image, however if the model hasn’t yet learned to generalize well because it didn’t see enough images \[2\] then the effect of LPL can be diluted in distributional metrics such as FID.
>
> Another way to see this effect is through the exposure bias problem [3].
>
> Broadly speaking, the effect of LPL is a polishing effect on the later reverse processes $p_t^\Theta, t> \tau_\sigma$.
>
> However, the exposure bias being significantly higher earlier during training ensures that the optimized distribution is practically not seen during sampling (Let $z_1 \sim \mathcal{N}(0, I)$ then $p_\Theta(z_t|z_1)$ is too different from $q(z_t|z_1)$ for LPL to have a meaningful effect early during training).
>
> Similarly, if the denoiser prediction of the latent is of very bad quality, then the gradient from LPL can be very noisy because of the non-linearities of the decoder dealing with out-of-distribution samples, thereby hampering convergence.
>
> Regarding \[1\], the method improves representation learning ability of the denoiser which can help it in better modelling object structures and semantic properties for example, but this does not address the latent-space/pixel-space mismatch tackled by LPL.
>
> Intuitively, it makes sense that such a method works better in pretraining, as it pushes the internal representation of the model towards a local minima induced by the pretrained teacher network, which works as a sort of knowledge distillation that is known to improve convergence properties.
>
> Also it will suffer less from the noisy gradient problem because their method does not require backpropagating through the teacher network.
>
>
> **2 -** We thank the reviewer for their remark, we will spend time improving the writing to better frame novelty, emphasize LDMs weaknesses and why this makes our LPL worthwhile.
>
> We thank the reviewer for their dedication in reviewing our work and providing valuable feedback and discussions.
>
> \[2\] GENERALIZATION IN DIFFUSION MODELS ARISES FROM GEOMETRY-ADAPTIVE HARMONIC REPRESENTATIONS, Kadkhodaie et. al, ICLR 2024.
>
> \[3\] M. Ning et al., Elucidating the Exposure Bias in Diffusion Models, ICLR 2024.

---

### Official Review · Reviewer_UNn8 · 2024-10-28

**Soundness:** 2
**Presentation:** 2
**Contribution:** 2
**Rating:** 5
**Confidence:** 4

**Summary:**

The author introduces latent perceptual loss which acts on the decoder's intermediate features to enrich the training signal of LDM. The experiments showcased the effect of the added perceptual loss.

**Strengths:**

- The author proposed latent perceptual loss (LPL) which shows the efficacy over various tasks datasets.
- The qualitative results and the quantitative metrics seem promising
- The frequency analysis showcases that the methods works
- The ablations are carried out in a systematic way

**Weaknesses:**

- The paper seems unpolished and rushed towards the deadline
- Sometimes the notation is a bit confusing, see the third point in the questions section
- Applying perceptual loss in T2I generation isn’t novel; Lin and Yang [1] calculates the perceptual loss in middle blocks to reduce computation which bypasses the computational constraint, whereas this paper requires passing results to the decoder for intermediate features. If the authors can provide evidence that the method proposed is better and distant themselves to the literature then I would consider raising my score.

---
[1] Lin, S., & Yang, X. (2023). Diffusion Model with Perceptual Loss. arXiv preprint arXiv:2401.00110.

**Questions:**

- Line 179, Many SOTA T2I diffusion models use a basic L2 training objective. Since only a VAE citation is given to explain blurriness, could you add citations or evidence that diffusion training leads to blurriness? Typically, adversarial losses are used in diffusion distillation, not in standard diffusion training.
- Line 185, "a perceptual loss cannot be used directly on the predicted latents", there is already accepted literature [2] that train another model which efficiently compute LPIPS losses in the latent space. It would be interesting to compare and contrast to the LatentLPIPS method.
- Line 193, it would be better if the authors could stick to the canonical DDPM/EDM [3] notations to avoid confusion.
- Line 289, do you enforce zero-terminal SNR here? It is known that the SD diffusion schedules are flawed [4].
---
[2] Kang, M., Zhang, R., Barnes, C., Paris, S., Kwak, S., Park, J., ... & Park, T. (2024). Distilling Diffusion Models into Conditional GANs. In ECCV 2024

[3] Karras, T., Aittala, M., Aila, T., & Laine, S. (2022). Elucidating the design space of diffusion-based generative models. Advances in neural information processing systems, 35, 26565-26577.

[4] Lin, S., Liu, B., Li, J., & Yang, X. (2024). Common diffusion noise schedules and sample steps are flawed. In Proceedings of the IEEE/CVF winter conference on applications of computer vision (pp. 5404-5411).

---

> ### Author Response · Authors · 2024-11-21
>
> ### Response UNn8
>
> We are glad the reviewer found our qualitative and quantitative results promising, and our ablations carried out in a systematic way. In the following, we respond to the points raised one by one.
>
> **W1+W2+Q3 “paper seems unpolished”, “the notation is a bit confusing”, “better if the authors could stick to the canonical DDPM/EDM \[3\] notations”**
> We updated our notation to be similar to the canonical notation as suggested, and proofread the text in order to improve its coherence and formatting. If the reviewer would like us to further clarify or otherwise improve specific passages of the manuscript we would be happy to do so.
>
>  **W2 “Applying perceptual loss in T2I generation isn’t novel, see Lin & Yang.”**
> We thank the reviewer for bringing this reference under our attention. We note that the paper by Lin & Yang \[1\] has not been peer-reviewed/published. However we added a discussion of this paper and compare with it experimentally in the new Section A.4 of the supplementary material. While Lin & Yang found their loss to deteriorate performance when using classifier-free guidance for sampling, we have observed consistently improved generations with LPL on different datasets and resolutions. For more detail we refer to Section A.4 of the supplementary material.
>
> **Q1 “only a VAE citation is given to explain blurriness, could you add citations…”**
> Another perspective is provided by Ning et al \[1\] , which attributes the smoothness/blurriness effect to exposure bias during sampling (denoiser input mismatch between training and sampling). We added a discussion of this work in the relevant work (L307-310).
>
> **Q2 “there is already accepted literature \[2\] that train another model which efficiently compute LPIPS losses in the latent space.”**
> We were referring to the usual LPIPS which employs a VGG-16 or AlexNet backbone trained in image space, such models are not suitable in our case.
> We compare LPL with E-LatentLPIPS as a training objective, results are reported in Table 5 of the appendix. We explore two scenarios: (1) training all the timesteps, and (2) training on only earlier timesteps. We observe that both approaches perform worse than the L2 baseline.
> Below, we summarize the main difference between our LPL and E-LatentLPIPS:
>
> * E-LatentLPIPS was only tested for distilling diffusion models into GANs and not for training diffusion models.
> * E-LatentLPIPS requires training of a classifier latent space for every new autoencoder, while LPL uses the already available autoencoder decoder.
> * In terms of interpretation, E-LatentLPIPS does not tackle the mismatch between pixel-space and latent-space diffusion but provides an LPIPS model that can help in latent distillation tasks. As this latent representation is more compressed than the latents, it is unlikely that it will help with high-frequency information similarly to ours.
> * It makes more sense that E-LatentLPIPS can help with later timesteps (high noise ratio) by providing semantic structures but this was not verified in our experiments.
> * When tested for diffusion training, we obtain results that are degraded with respect to the L2 baseline.
>
> **Q3:** addressed above
>
> **Q4 “do you enforce zero-terminal SNR here? It is known that the SD diffusion schedules are flawed \[4\].”**
> Yes, our velocity models (DDPM-v and flow) all use a zero terminal SNR state, we now state this in lines 300-302.
>
> \[1\] M. Ning et al., Elucidating the Exposure Bias in Diffusion Models, ICLR 2024
> \[2\] Kang et al., Distilling Diffusion Models into Conditional GANs. In ECCV 2024

---

> > ### Comment · Reviewer_UNn8 · 2024-11-26
> >
> > Thank you for the effort and thought you have put into this rebuttal. I sincerely appreciate the additional analysis and insights you have provided.
> >
> > **W2**: Thank you for the additional results with respect to [1], and the investigation that went into this. With the new tables and explanations, I agree that LPL demonstrates stronger performance in this context. I encourage you to incorporate these results and corresponding discussions into the main paper.
> >
> > **Q2**: As noted by myself and the other reviewers, the claims regarding novelty should be reconsidered in light of prior work. That said, I value the additional experiments provided in the rebuttal. I encourage you to also integrate these experiments and discussions into the final version of the paper. With regard to the authors' third claim
> >
> > > As this latent representation is more compressed than the latents, it is unlikely that it will help with high-frequency information similarly to ours.
> >
> > The experimental results presented in [2], specifically Fig. 5, provide clear evidence that E-latentLPIPS is indeed capable of capturing high-frequency details.
> >
> > Overall, the rebuttal has clarified some of my points. While I believe there is still room to refine the framing of novelty, I appreciate the improvements made and will improve my score accordingly.
> >
> > ---
> > [1] Lin, S., & Yang, X. (2023). Diffusion Model with Perceptual Loss. arXiv preprint arXiv:2401.00110.
> >
> > [2] Kang, M., Zhang, R., Barnes, C., Paris, S., Kwak, S., Park, J., ... & Park, T. (2024). Distilling Diffusion Models into Conditional GANs. In ECCV 2024

---

> > > ### Author Response · Authors · 2024-12-04
> > >
> > > We thank the reviewer for their valuable feedback.
> > >
> > > **W2 - Q2.**
> > > We thank the reviewer for their suggestions, we will incorporate these results into the main manuscript and revisit any novelty claims in light of the related works.
> > >
> > > **Q2’.**
> > > We thank the reviewer for pointing out to this Figure, it seems that E-latentLPIPS is capable of improving modelling of high level details in the case of GAN distillation.
> > >
> > > In our experiments training diffusion models, we observed E-latentLPIPS to lead to smooth/blurry images, however we will not make any claims regarding this as we did not perform quantitative experiments to assess this effect.
> > >
> > > Thank you again for taking the time to review our paper and for providing constructive feedback!

---

### Official Review · Reviewer_iXeA · 2024-11-02

**Soundness:** 3
**Presentation:** 3
**Contribution:** 3
**Rating:** 8
**Confidence:** 4

**Summary:**

The paper proposes applying perceptual losses to latent diffusion models during training to improve the quality of generated images. Their proposed perceptual loss uses features from the VAE decoder, with additional contributions such as outlier filtering being proposed to enable this setup to work. The authors extensively validate the benefits of their approach over standard training of MMDiTs in different diffusion formulations and datasets.

**Strengths:**

The comparisons with baselines are extensive and clearly show that the proposed latent perceptual loss objective improves metrics across different diffusion formulations and datasets. I really appreciate the authors demonstrating the improvements for both eps-pred diffusion and flow matching, demonstrating that the proposed loss potentially has a general significance.

The paper ablates over/explores a large number of parameters and their influence.

The paper is generally reasonably well-written and accessible.

**Weaknesses:**

I think framing the proposed loss as a perceptual loss is likely incorrect. Perceptual losses typically try to incorporate human perception-based invariances into the loss, such as weighting the presence of the correct texture (e.g., grass) as more important than getting every detail of the instance of the texture (e.g., the exact positions of individual blades of grass) right. This is directly opposed to losses such as the MSE in pixel space. This is typically accomplished by taking features of a deep pre-trained discriminative model. In this case, you use features of the VAE latent decoder, which, potentially even puts the features at a lower abstraction level than the original latents. As far as I could see, there was no investigation of whether these features have the qualities of a perceptual loss. In the end, optimizing this loss seems to result in improved FIDs, which makes it a valuable contribution. Still, I think calling it a perceptual loss without further investigation goes against what people expect "perceptual losses" to refer to, which could lead to confusion. Could it be that the improvement actually doesn't come from perceptual qualities of the loss but rather from other qualities, such as a different implicit weighting of timesteps, which has previously been shown to substantially improve FIDs as well [https://openaccess.thecvf.com/content/ICCV2023/papers/Hang_Efficient_Diffusion_Training_via_Min-SNR_Weighting_Strategy_ICCV_2023_paper.pdf].

You also claim that "the autoencoder’s latent space has a highly irregular structure and is not equally influenced by the different pixels in the latent code" as an important part of your motivation. However, it has been shown [e.g., in https://discuss.huggingface.co/t/decoding-latents-to-rgb-without-upscaling/23204] that, at least for the very commonly used SD VAEs, the autoencoder latents effectively correspond to downsampled colors, differing fundamentally from that statement. Similarly, noisy latents from intermediate steps of the sampling process often already show a good approximation of the image being generated. A.2 shows an investigation of the effect of spatial interpolation for some (unspecified) AE, but is inherently very limited in its scope. It would be nice to see better backing up of this central claim that goes against common assumptions using standard methods, such as introducing small (random) deviations and showing how their effect on the decoded image is disproportionally large compared to larger ones. General claims should also be verified with standard VAEs.

**Questions:**

In addition to the addressing the weaknesses, I'd appreciate the authors answering the following questions:

What is the difference in training FLOPS per step of using LPL vs. just the normal objective? I assume it is a minor difference, but if it is not, just comparing to normal objectives at the same step count would not be fair. Similarly, the effect of the substantial increase in VRAM usage on the training speed (via the maximum achievable local batch size) should be quantified properly.

Why do you use a CFG scale of 2? Standard choices for latent diffusion are typically much lower for ImageNet class-conditional generation and much higher for text-to-image.

Why is the frequency cut off at 360 in Fig. 6?

Suggestion: while not technically required according to ICLR's rules, as it hasn't been published yet (and therefore also not taking an influence on my rating of this paper), I think a discussion of the almost one year old seminal work [https://arxiv.org/abs/2401.00110] in the area of applying perceptual losses to diffusion models would help improve the paper, especially if this work had any influence on the submitted paper.

---

> ### Author Response · Authors · 2024-11-21
>
> ### Response iXeA
>
> We were happy to see that the reviewer states that our extensive experiments “clearly show that the proposed latent perceptual loss objective improves metrics across different diffusion formulations and datasets”.
> We respond to the different points raised by the reviewer point-by-point below.
>
> **W1 “framing the proposed loss as a perceptual loss”**
> Our motivation to use the term perceptual is the conceptual similarity to other losses based on deep features, which are typically based on classifiers. The Self-Perceptual approach of \[1\] Lin & Yang, is also not based on classifier features. Our motivation to name our approach Latent Perceptual Loss allows us to make the connection to this related work, rather than to claim a connection between our work and human perception. Note that other “perceptual metrics”, such as DreamSim \[2\], rely on deep features from non-discriminative networks such as DINO and CLIP, see Section 4 of \[2\]. We can discuss this point in the introduction to avoid any confusion on this point if the reviewer feels this would be helpful.
>
> **W2 “Could it be that the improvement actually… \[comes from \] … different implicit weighting of timesteps”**
> Thank you for this interesting question, we address it in the newly added Section A.5 of the supplementary material. In Table 6, we verify that it is our Latent Perceptual Loss (LPL) rather than timestep weighting that underlies the performance improvements by comparing it to two different baselines that do not use LPL: (a) reweighting the loss of later timesteps to be similar to the implicit reweighting induced by using our LPL for later timesteps, (b) the state-of-the-art reweighting technique of Hang et al. \[3\]. We find that neither  of these baselines brings the improvements over the vanilla epsilon weighting, and that the best performing approach is our LPL. Thus, we conclude that the hypothesis that the improvements are not coming from LPL is unlikely.
>
> **W3 “It would be nice to see better backing up… such as introducing small (random) deviations and showing how their effect on the decoded image is disproportionally large compared to larger ones”**
>
> Similar to the discussion on the huggingface forum cited by the reviewer, we can fit a linear projection to directly predict the downscaled images from the latents, and then compare the decoding results. Please keep in mind that such a linear decoding produces images 8x smaller than the original, hence many details are lost. We provide two new experiments in section A.2 of the appendix, see paragraph “Linearity of the latents”.
> First, we find that the linear decoding idea from the cited discussion does not work well: the optimal linear mapping varies with the input image, and they introduce both high-frequency noise artifacts, as well as low-frequency color tone mismatches.
> Additionally, such reconstructions are very approximative and are 8x smaller than the original image, hence the details can not accurately be represented with this method.
>
> Finally, regarding  our claim that the AE latent is “highly irregular”, we view this as a consequence of the compression rate of the autoencoder which we intuit relies on high frequencies to encode high level information.
> Our experiment in section A.2, in which the interpolation operation simulates a low-pass filter on the latents, showcases this hypothesis in the case of SD autoencoders.
> When taking into consideration the tendency of l2 to produce blurry results on regression tasks, we can conclude that the training objective is not ideal for the nature of the latent space.
> We are open to accommodate reviewer’s suggestions on how to soften this claim based on the provided evidence.

---

> ### Author Response · Authors · 2024-11-21
>
> **Q1 FLOPS and VRAM usage**
> We have added a new section A.6 to the supplementary material to study the effect of the computational and memory overheads induced by our Latent Perceptual Loss (LPL).
> Results in Table 7 show that when maximizing the batch size for the available memory, and training for the same wallclock time of 48h, LPL still achieves improved FID scores despite a reduced throughput. A detailed tracking of FID over training iterations in Fig 15 shows that LPL leads to consistent improvements over the baseline after as little as 20k iterations.
>
> **Q2 “Why do you use a CFG scale of 2?“**
> We follow the standard approaches when choosing the guidance scale, a value 2 of is a defacto standard when comparing ImageNet models at 512 resolution, at 256 resolution we use a value of 1.375 which is the standard (such values give the best FID results for the baseline) \[4,5\], higher values can produce more visually appealing images but at the cost of reduced diversity.
>
> **Q3 “Why is the frequency cut off at 360 in Fig. 6?”**
> For the frequency graph, we plot the errors for different frequency ranges, for images that are 512x512. The frequency range corresponds to circles with radius between 0 and the maximum radius in the image, i.e., the distance from the center to one of the corners, this distance is given by sqrt(256\*\*2 \+ 256\*\*2) \= 362\.
>
> **Q4 Suggested discussion of Lin & Yang, arXiv 2023 \[6\]**
> We thank the reviewer for this suggestion, we discuss this paper and compare with it experimentally in the new Section A.4 of the supplementary material. While \[6\] found their loss to deteriorate performance when using classifier-free guidance for sampling, we have observed consistently improved generations with LPL on different dataset, resolutions, and datasets. For more detail we refer to Section A.4 of the supplementary material.
>
>
> \[1\] Self-Perceptual, Lin & Yang 2024, https://arxiv.org/abs/2401.00110.
> \[2\] DreamSim: Learning New Dimensions of Human Visual Similarity using Synthetic Data, Fu et al., NeurIPS 2023\. https://openreview.net/forum?id=DEiNSfh1k7
> \[3\] Efficient Diffusion Training via Min-SNR Weighting Strategy, Hang et al., ECCV 2023\.
> \[4\] Scalable Diffusion Models with Transformers, Peebles & Xie, ICCV 2023\.
> \[5\] On Improved Conditioning Mechanisms and Pre-training Strategies for Diffusion Models, Berrada et. al., NeurIPS 2024
> \[6\] Diffusion Model with Perceptual Loss. Lin & Yang, arXiv 2023\.

---

> > ### Comment · Reviewer_iXeA · 2024-11-24
> >
> > Thank you for the detailed response and extensive additional material provided. I appreciate the effort that went into this.
> >
> > **W1**: What all of the examples of typical perceptual losses mentioned have in common is that these features can reasonably be expected to be at a higher level of abstraction than the original, one where deviations in feature space align with perceptual deviations in image space. Getting such features from a (V)AE *decoder* is unintuitive. Without a investigation as to whether these features constitute a perceptual loss, I stand by my initial statement in this regard. Similarly to W3EV, I consider this to be more heuristic than principled. However, a contention about semantics is not particularly relevant to the paper's main contribution, where I think the practical improvements are still valuable regardless of semantics.
> >
> > **W2**: Thank you for the additional investigation. I think, optimally, one would equalize by the contribution of the loss to the gradient instead of variances, but the given experiment is sufficient to resolve my concerns.
> >
> > **W3**: Thank you for the experiment. However, contrary to your interpretation, I think it shows exactly what I was referring to: in the last column, the images reconstructed from the linear projection very closely resemble the input images. Some noise and color shift are present, but the general image is there. In my mind, the fact that a single linear probe trained on a single image can recover the majority of the image across multiple samples shows exactly that the space you are operating on is *not* "highly irregular".
> >
> > **Q1**: Thank you for adding that comparison, this resolves my concern. You might want to consider incorporating it in the main body of the paper.
> >
> > **Q2**:  I'm not aware of many papers using a CFG scale of 2 on ImageNet-512 -- DiT (1.5), EDM2 (1.4), U-ViT (1.7), DiffiT (1.5), or simple diffusion (1.1), to name a few important works, all use lower CFG scales for optimal FID values, as mentioned.
> >
> > **Q3**: Thank you for the explanation. I agree, this makes sense.
> >
> > **Q4**: Thank you for the additional comparisons, my concerns have been resolved. Please consider adding these discussions and comparisons to the main paper, I think they would significantly improve the paper.

---

> > > ### Author Response · Authors · 2024-11-26
> > >
> > > We thank the reviewer for their thoughtful feedback.
> > > **W1 “ Without an investigation as to whether these features constitute a perceptual loss, I stand by my initial statement in this regard.”**
> > > If the reviewer views this point as important, we are willing to change the name of our method to something different like *LPL \- Latent Pixel Loss* or *LDFM : Latent Decoder Feature Matching*.
> > >
> > > **W3 “... shows exactly that the space you are operating on is not "highly irregular"”**
> > > We agree that the use of “highly irregular” is an overstatement, we therefore tone down this claim in the manuscript, replacing it with “non-uniform”.
> > > Additionally, we provide an additional experiment supporting the claim that some regions in the latent space can have more impact on the reconstructed image than others.
> > > Illustrated in Figure 12, where we show that perturbing certain regions in the latent space by adding small amounts of gaussian noise can induce errors globally in the image while others have a localized effect, and that the error amplitude also varies significantly with the choice of the region (l857-863).
> > > We hope that the following changes address the reviewer’s concern.
> > >
> > > **Q2 “I'm not aware of many papers using a CFG scale of 2 on ImageNet-512”**
> > > Regarding FID, we are most closely related to DiT which chooses a guidance scale of 1.5 for 256 resolution, but their method uses only 3 channels for the guidance and they show that this is equivalent to using the four channels for the CFG with a scale of 1.375.
> > > We provide a sweep over different CFG values in Figure 19, showing the optimal FID for our baseline at 512 resolution is obtained with a guidance scale of 2.0.
> > >
> > > **Q4 “... Please consider adding these discussions and comparisons to the main paper, I think they would significantly improve the paper”**
> > > Thank you for your suggestion, we are adding these discussions to the main manuscript.
> > > We will continue improving the storyline and the flow of the main paper accordingly.

---

> > > > ### Comment · Reviewer_iXeA · 2024-12-01
> > > >
> > > > Thank you for the response and additional information.
> > > >
> > > > Regarding your comments:
> > > >
> > > > **W3** & **Q2**: Thank you for the additional experiments and clarification. This resolves my concerns.
> > > >
> > > > **W1**: I would only like to ask the authors to consider what they think the most appropriate term for their work is. As stated previously, I think changing the term could be beneficial, but I won't let semantics questions influence my rating. Please feel free to choose the terminology you find most appropriate.
> > > >
> > > > Overall, the answers provided by the authors and the extensive additional information and experiments provided during the rebuttal substantially improved the submission. While I agree with W3EV that the method seems somewhat heuristic, the authors have extensively demonstrated their approach's practical merit in various settings, making them a valuable contribution. Especially the additional experiments presented during the discussion phase such as Tab. 6 & 7, make a compelling argument for the method's relevance. Therefore, I'm going to increase my score and hope for acceptance.

---

> > > > > ### Author Response · Authors · 2024-12-04
> > > > >
> > > > > We thank the reviewer for their valuable feedback.
> > > > >
> > > > > **W1.** Thank you for the clarification, we will keep the term Latent Perceptual Loss but will make sure to state in the manuscript that our use of the term “Perceptual” diverges from the traditional sense because we did not perform experiments showcasing perceptual qualities of the decoder.
> > > > >
> > > > > However, in our case the use of the term “*perception*” relates to how errors in latent space are perceived in image space.
> > > > >
> > > > > We are very grateful for the reviewer’s hard work on assessing our paper and thankful for them raising their score.

---

### Official Review · Reviewer_sAd7 · 2024-11-05

**Soundness:** 3
**Presentation:** 3
**Contribution:** 2
**Rating:** 5
**Confidence:** 4

**Summary:**

The paper studies the perceptual loss function in the training of diffusion models and proposes to compare the features between $z_0$ and $\hat{z}_0$ by sending them into the latent autoencoder's decoder model. The loss is only applied during the post-training stage and only applied in the time steps that have a SNR higher than the preset threshold. The results on CC3M, ImageNet-1k and S320M show that the introduction of this training strategy could help improve the generation quality of the model.

**Strengths:**

1. The motivation is clear and straightforward. And the proposed method is simple and can be easily applied to the training of other diffusion models.
2. Under the same training iterations during the post-training stage, the method can improve the FID over the baseline method which only adopts the MSE loss.

**Weaknesses:**

1. The paper only shows the performance increase over the baseline model. I feel like it's better to clearly demonstrate the effectiveness and performance gain over the previous state-of-the-art methods, to show that the perceptual loss can achieve what the widely used MSE loss cannot achieve.
2. The introduction of the perceptual loss would increase the computation cost during the training stage. Could the authors provide a clear comparison on this?
3. Following the last one, what would be the performance comparison for the same training time instead of the same training iterations?
4. The authors mention the outliers in the features of the autoencoder's decoder which is not ideal for the computation of the perceptual loss. I'm wondering if the authors have tried other ways instead of simply masking those features out as this might cause information lost. Or has the authors tried using some other models to compute the perceptual loss to avoid those outliers?

**Questions:**

See weaknesses.

---

> ### Author Response · Authors · 2024-11-21
>
> ### Response sAd7
>
> We thank the reviewer for their encouraging comments: “... motivation is clear …” and “method is simple and can be easily applied”. Below we address the different points raised in the review one-by-one.
>
> **W1 “clearly demonstrate the effectiveness and performance gain over the previous state-of-the-art methods”**
> Our baseline is based on the state-of-the-art SD3: we replicate their multimodal DiT (MMDiT) architecture and flow matching training procedure, and train the model on two public datasets (for reproducibility) as well as one proprietary dataset. Thus, we argue that the reported results should be considered as improvements over state-of-the-art methods. To better show that the baseline models we trained are competitive with SotA models, we provide a quantitative comparison on ImageNet at 512 resolution in Table 8 in the new Section A.7 of the supplementary material, showing that our Flow model, using RK4 ODE sampler already achieves SoTA results in terms of FID.
>
> **W2+W3 “perceptual loss would increase the computation cost … provide a clear comparison on this”, “performance comparison for the same training time instead of the same training iterations”**
> We thank the reviewer for this suggestion. We have added a new section A.6 to the supplementary material to study the effect of the computational and memory overheads induced by our Latent Perceptual Loss (LPL).
> Results in Table 7 show that when maximizing the batch size for the available memory, and training for the same wallclock time of 48h, LPL still achieves improved FID scores despite a reduced throughput. A detailed tracking of FID over training iterations in Fig 15 shows that LPL leads to consistent improvements over the baseline after as little as 20k iterations.
>
> **W4 “wondering if the authors have tried other ways instead of simply masking … Or has the authors tried using some other models to compute the perceptual loss to avoid those outliers?”**
> (a) In our initial experiments we tried other approaches than masking, e.g. training the AE with L1 loss and clipping large activations. However, none of these approaches worked as well as the introduced outlier detection approach.
>
> (b) We found outlier features in both AE architectures we explored: the asymmetric autoencoder from Zhu et al. (2023) and the autoencoder from SDXL (Podell et al., 2024). Recent papers (Kang et al. ECCV’24, Lin & Yang arXiv’24) consider other models to compute a perceptual loss: an image classifier train in autoencoder latent space, and a diffusion model Unet features, respectively. In the newly added Section A.4 of the supplementary material we provide a discussion-of and comparison-to these approaches. Experimentally we find LPL to lead to better results than these alternative approaches.

---

> > ### Author Response · Authors · 2024-11-28
> >
> > Dear reviewer, we updated our manuscript with additional experiments regarding the computational overhead (**W2+W3**).
> > * Table 8 summarizes the total increase in FLOP count during training when accounting for the addition of our method, for the different experiments in Table 2.
> > * We find the total FLOP count in every case to only increase by (10-15%) compared to the baseline.
> > * Figures 17 and 18 transpose Figure 15 to the case where the x-axis represents the total FLOP count.
> > * Under this setting, our method outperforms the baseline even when running it for a longer number of iterations in order to equalize the FLOP count.
> >
> > We hope that the additional experiments address your concerns.
> > Please feel free to share your point-of-view on our rebuttal and any other points that you would like us to further clarify.

---

### Author Response · Authors · 2024-11-21

# Global Response

We thank the reviewers for their insightful remarks on our work, and encouraging comments on the effectiveness of our method and the extensiveness of our experiments and ablations.

We have updated the manuscript following the suggestions in the reviews, and marked the additions in blue to facilitate finding these passages.

In the following we respond to three main critiques raised by the reviewers: Comparison with existing works employing perceptual losses for diffusion models; efficiency of our method; and novelty of our method.

## Additional related work. (iXeA, UNn8, W3EV)

Thank you for bringing \[1,2\] additional references under our attention. Although \[1\] hasn’t been published yet and we consider \[2\] as concurrent to our work (the ICLR submission deadline coincided with the ECCV conference), we believe that discussing these papers in related work and providing experimental comparisons would strengthen our paper.

**\[1\] Self-Perceptual, Lin & Yang 2024,** https://arxiv.org/abs/2401.00110. This method has been devised on UNet denoiser architectures and the main idea is to use a copy of the denoiser network itself as a feature extraction network on which to compute the perceptual loss. We note however that this method did not improve results when compared with training using the standard L2 objective with classifier-free guidance applied for sampling (cf. Table 8 caption: ”Combining our self-perceptual objective with classifier-free guidance does improve sample quality but does not surpass the MSE objective with classifier-free guidance.") Furthermore, it is unclear how to adapt such a method to SotA DiT-based architectures. Lin et al. choose the middle layer of the UNet, which is the most semantic and lowest resolution, for DiT-based architectures there is no notion of downsampling, we experiment with implementing this method and using different layer depths (8, 16 and 28\) in our experiments. Finally, Lin et al. apply the Self-Perceptual loss throughout the entire training which poses significantly more overhead than our method which is only used during finetuning.

**\[2\] Latent-LPIPS, Kang et al., ECCV’24,** https://arxiv.org/abs/2405.05967. This work proposes a variant of the LPIPS loss based on image classifiers trained in the latent space of different autoencoders. The authors did not directly train a diffusion model with this loss, but instead used it to distill a pretrained diffusion model into a GAN. Using latent-space classifiers released by Kang et al., we experiment with Latent-LPIPS of this paper, using it either on all timesteps or for earlier timesteps only (as we do in our lpl method).

We added a Section A.4 to the appendix to discuss these related works and compare with them experimentally. In Table 5 we report our experimental results in terms of FID training throughput, and memory usage, and find that our Latent Perceptual Loss yields better results, and moreover that neither of these approaches improves over our DiT-based baseline model.

## FLOPS and training speed. (sAd7, iXeA, W3EV)

We agree with the reviewers that it would be useful to provide more details on the training speed and memory requirements. The newly added sections A.4, A.5 and A.6 to the supplementary material to provide more insight here.
(1) in Table 5, we compare memory usage and throughput of different methods; (2) in Table 7, we conduct a comparison where both the baseline and our model are trained with the maximum batch size that fits into memory for a period of 48 hours; and (3) in Figures 14 and 15, because our method only needs to be applied during finetuning, we provide a finegrained study of the performance improvements where FID is computed after every 10k iterations on a converged ImageNet model. In all the three cases, we observe superior performance w.r.t the considered baselines.

## Novelty (W3EV).

The novelty of our method comes from identifying an important problem in latent generative models –mismatch between pixel diffusion and latent diffusion–,   and providing a simple and efficient solution that works in practice. In our experiments, we observe boosts between 6% and 20% in FID w.r.t. state-of-the-art architectures such as SD3. To ensure the robustness of our observations, we perform extensive experiments and ablations showcasing the importance of each component of our LPL. Thus, we believe that our manuscript provides a valuable contribution to the field. Additionally, to strengthen our contribution, we provide a derivation of the LPL objective from a point of view of pixel level penalty on the forward process posterior in diffusion and show that it theoretically holds for small timesteps where the predicted latent is similar to the original latent (see A.1 in the updated manuscript).

---

### Meta-Review · Area_Chair_1nEU · 2024-12-19

**Metareview:**

Summary: Proposed a perceptual loss in the context of latent diffusion models to improve image sharpness. It uses features extracted from the latent decoder to calculate the distance between z0 and the predicted z0_hat at the pyramid levels of the decoder. Demonstrates improved image quality compared to standard baselines, as shown on different models and dataset settings.

Strength: The paper is motivated well. The proposed approach is easy to understand. The technique is simple to implement and incorporate into existing LDM training loops. The paper shows clear improvements in image quality over the baseline. Similarly, the ablations are strong, for instance visualizing the presence of more high frequency signals when the proposed perceptual loss is employed.
Acceptance Reason: The paper identified an important problem in LDM models, where their generations could be blurry. It’s a simple and easy to implement solution that is proven to enhance image quality, and I believe this is a good contribution to the community.

**Additional Comments On Reviewer Discussion:**

The paper received 1x accept, 1x marginally above acceptance and 2x marginally below acceptance. Concerns raised on writing have been adequately addressed. Novelty also raised as a common concern among the reviewers. For instance, reviewer UNn8, pointed to a paper that proposes a related approach. However, the referred paper is not peer-reviewed or published. For the most part, the rebuttal has addressed most of the questions, with the exception of reviewer (sAd7).

---

### Decision · Program_Chairs · 2025-01-22

Accept (Poster)